# Endothelial cell heterogeneity and microglia regulons revealed by a pig cell landscape at single-cell level

Fei Wang[1,2,3,21], Peiwen Ding [3,4,21], Xue Liang[1,5,21], Xiangning Ding[3,4,21], Camilla Blunk Brandt [2,6,21], Evelina Sjöstedt[7], Jiacheng Zhu[3,4], Saga Bolund[7], Lijing Zhang[3,4,8], Laura P. M. H. de Rooij[9,10], Lihua Luo[3,4], Yanan Wei[3,11], Wandong Zhao[3,11], Zhiyuan Lv[3,11], János Haskó[2], Runchu Li[3,11], Qiuyu Qin [3,11], Yi Jia[3,11], Wendi Wu[3,11], Yuting Yuan[12], Mingyi Pu[3,11], Haoyu Wang[3,4], Aiping Wu[13,14], Lin Xie[8], Ping Liu[8], Fang Chen[8], Jacqueline Herold[2], Joanna Kalucka [2,6,15], Max Karlsson [16], Xiuqing Zhang[3,4], Rikke Bek Helmig[17], Linn Fagerberg [16], Cecilia Lindskog [18], Fredrik Pontén [18], Mathias Uhlen [7,16], Lars Bolund[1,2], Niels Jessen [6], Hui Jiang[8], Xun Xu [3], Huanming Yang[3,19], Peter Carmeliet [2,9,10,20], Jan Mulder [7], Dongsheng Chen[3,13,14✉], Lin Lin [2,6✉] & Yonglun Luo [1,2,3,6,19✉]

Pigs are valuable large animal models for biomedical and genetic research, but insights into the tissue- and cell-type-specific transcriptome and heterogeneity remain limited. By leveraging single-cell RNA sequencing, we generate a multiple-organ single-cell transcriptomic map containing over 200,000 pig cells from 20 tissues/organs. We comprehensively characterize the heterogeneity of cells in tissues and identify 234 cell clusters, representing 58 major cell types. In-depth integrative analysis of endothelial cells reveals a high degree of heterogeneity. We identify several functionally distinct endothelial cell phenotypes, including an endothelial to mesenchymal transition subtype in adipose tissues. Intercellular communication analysis predicts tissue- and cell type-specific crosstalk between endothelial cells and other cell types through the *VEGF*, *PDGF*, *TGF-β*, and *BMP* pathways. Regulon analysis of single-cell transcriptome of microglia in pig and 12 other species further identifies *MEF2C* as an evolutionarily conserved regulon in the microglia. Our work describes the landscape of single-cell transcriptomes within diverse pig organs and identifies the heterogeneity of endothelial cells and evolutionarily conserved regulon in microglia.

---

A full list of author affiliations appears at the end of the paper.

The domestic pig (*Sus scrofa domesticus*) is an important large animal for modeling both monogenic and complex human diseases such as cystic fibrosis[1], atherosclerosis[2], Huntington's diseases[3], and diabetes[4–6]. Additionally, it is explored and regarded as the most promising source for xenotransplantation[7–9] largely due to the postulation of their high resemblance to humans in organ size, structure, anatomy, genetics, and physiological functions[10,11]. Recently, clinical transplantation of genetically modified pig kidneys and heart has successfully been achieved with patients[12]. Despite all these great promises and progress, several species-specific cellular and molecular differences between pigs and humans exist. For example, pluripotency progression and metabolic transition were found to be different using single-cell RNA-sequencing (scRNA-seq)[13]. To gain better insights into the biomedical similarity, as well as differences between pig and human, and to advance the applications of pigs in biomedical research, a comprehensive and body-wide investigation of the domestic pig is needed at single-cell level.

High-throughput scRNA-seq technology has greatly expanded the ability to better understand the cell composition, interactions, heterogeneity, and functions in highly organized and multi-cellular mammalian organs under physiological or pathological conditions[14,15]. Pioneering work has been completed for most model animals and humans[16–22]. Since the completion of the first porcine reference genome, technological breakthroughs in gene editing and cloning allow precise manipulation of the pig genome in living animals[23–25] and applications of pigs in biomedical research have been greatly expanded. Previous scRNA-seq studies in pigs were restricted to a few tissues or a single organ[26–28]. Recently, the pig BodyMap atlas with the conventional bulk RNA-seq method has been reported[29]. Using similar bulk RNA-seq based transcriptome profiling, we have systematically profiled the protein-coding transcriptome in 98 pig tissues (www.rnaatlas.org)[30].

In this study, we report the generation of a single-cell transcriptome atlas of 222,526 cells across twenty tissues of the domestic pig by using scRNA-seq and single-nuclei RNA-sequencing (snRNA-seq). The mapping results are available through the pig single-cell atlas database for comparative analyses and data exploration (https://dreamapp.biomed.au.dk/pigatlas/). In total, 58 cell types were identified, which contribute to tissue-specific and shared functions between the tissues. We identified tissue-specific cell types, as well as common cell types shared across different tissues. Commonly shared cell types also exhibit tissue-specific expression patterns and functions. One such cell type is vascular endothelial cells (ECs). Further analysis of ECs probed rare ECs types supporting the notion of endothelial to mesenchymal transition (EndMT), which we further validated by scRNA-seq of cultured ECs and induced EndMT by transforming growth factor-beta 2 (TGF-β2) treatment. These single-cell transcriptome data allow us to gain insights into the similarities and differences in biomedical and cellular functions between pigs and humans. We also performed a pan-species regulon comparison analysis covering thirteen different species and identified *MEF2C* as the most conserved regulon for microglia evolution over 300 million years.

## Results

### Single-cell and single-nuclei RNA sequencing of four pig tissues highly correlates in common cell types.
To construct the first body-wide single-cell transcriptome atlas of pigs, 20 pig tissues/organs were analyzed using single-cell and/or single-nuclei sequencing. Nine pig tissues, including visceral and subcutaneous adipose tissues, spleen, intestine, liver, lung, peripheral blood mononuclear cells (PBMCs), whole brain, and retina, were analyzed by single-cell RNA sequencing (scRNA-seq). Fifteen tissues, including nine brain regions, retina, kidney, heart, spleen, liver, and lung, were analyzed by single-nuclei RNA sequencing (snRNA-seq). Among the nine different brain regions, four regions (area postrema, cerebellum, subfornical organ, and the vascular organ of lamina terminalis (OVoLT)) were generated by this study, and the other five brain regions (frontal lobe, hypothalamus, occipital lobe, parietal lobe, and temporal lobe) were from a study we had previously published using snRNA-seq[28]. Both scRNA-seq and snRNA-seq techniques were used for single-cell transcriptome analysis, which had both technical strengths and limitations[31]. The scRNA-seq is performed using freshly isolated single cells, thus capturing all transcripts in the cells, but limited by the sample processing procedures. Single-cell suspensions must be prepared from fresh tissues for scRNA-seq. For snRNA-seq, tissues can be snap-frozen after sampling and used for nuclei extraction, thus less limited by timing. We selected both scRNA-seq and snRNA-seq for reasons of practicality and resource availability. To compare the two methods, four pig tissues (liver, retina, lung, spleen) were analyzed with both scRNA-seq and snRNA-seq. In total, the pig single-cell transcriptome atlas includes twenty pig tissues (Fig. 1a). All samples were dissociated into single cells or single-nuclei, followed by high-throughput scRNA-seq or/and snRNA-seq library generation, deep sequencing, and data analyses (Fig. 1b). Complete information of all batches of samples was summarized in Supplementary Data 1. After filtering low-quality cells (nFeatures_RNA < 200 or % mitochondria transcripts > 30%) and doublets (nFeatures_RNA > 5000), high quality single-cell transcriptome data were obtained from 222,526 cells, of which 133,492 and 89,034 were obtained with scRNA-seq and snRNA-seq, respectively.

We first compared single-cell gene expression profiling of tissues obtained with scRNA-seq and snRNA-seq. Quality control (QC) comparison of scRNA-seq and snRNA-seq demonstrated that scRNA-seq captures more transcripts per cell and obtains higher percentage of mitochondria- and ribosome genes. In contrast, snRNA-seq captures a higher percentage of protein-coding transcripts and transcription factor-encoding gene transcripts, largely due to the absence of mitochondria and ribosome RNAs. Despite that, when evaluating cell cycle gene expression, the scores of S and G2M stages are similar in tissues analyzed with scRNA-seq and snRNA-seq (Supplementary Data 2). Next, cells from the spleen, liver, lung, and retina captured by snRNA-seq and scRNA-seq were clustered and visualized by t-SNE plots (Supplementary Fig. S1). Cell clusters were annotated based on the expression of canonical cell-type markers (Supplementary Fig. S1a–d). The number captured cells for each cell type in the four shared tissues analyzed by scRNA-seq and snRNA-seq are highly variable (Supplementary Fig. S1e–h), which can be explained by sampling and tissue processing biases, as the scRNA-seq and snRNA-seq experiments were carried out in two different laboratories. Despite that, our results showed that common cell types captured by scRNA-seq and snRNA-seq exhibit similar gene expression profiles and cell-type-specific markers. Most importantly, correlation analysis between shared cell types further showed that the transcriptome of the same cell types identified scRNA-seq and snRNA-seq are well correlated (Supplementary Fig. S1i–l). Thus, we combined the scRNA-seq and snRNA-seq data, batch-corrected, and performed an integrative analysis with *Seurat*[32].

### A pig single-cell transcriptome atlas.
To identify the cell types based on their single-cell transcription profile, we first performed graph-based clustering and visualized all the cells from these twenty tissues using t-distributed stochastic neighbor embedding

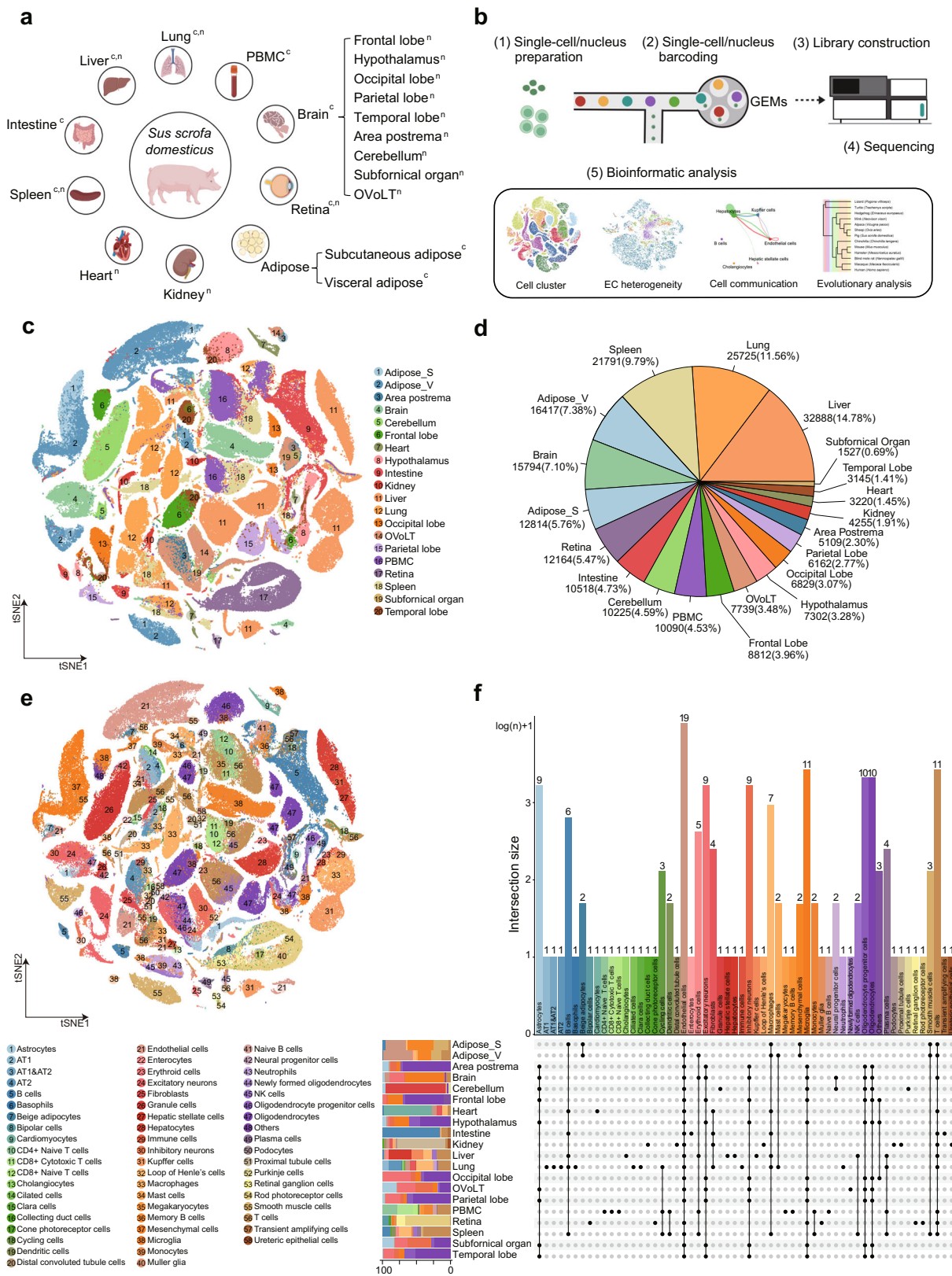

(t-SNE) at tissue levels (Fig. 1c). On average, the number of single-cell/nuclei transcriptomes obtained from each tissue ranges from 1527 cells (subfornical organ) to 32,888 cells (liver), representing 0.69% and 14.78% of the total cells, respectively (Fig. 1d). Most cells from each tissue are clustered separately thus

corroborating the general transcriptome regulation of tissue origin[33]. We next performed cell-type clustering and annotated each cluster of cells according to the expression of canonical cell-type markers for each tissue separately (Supplementary Data 3). In total, we identified 234 cell clusters corresponding to 58 major

**Fig. 1 A single-cell transcriptome atlas of 20 pig tissues. a** Schematic diagram of organs/tissues. Superscripts "c" and "n" represent the tissue analyzed by scRNA-seq and snRNA-seq respectively. OVoLT, vascular organ of lamina terminalis. **b** Schematic diagram of cDNA libraries generation and downstream bioinformatics analyses. The scRNA-seq and snRNA-seq were constructed independently, followed by high-throughput sequencing, and downstream bioinformatic analyses. **c** t-SNE visualization of all single cells in the 20 tissues. Cells are color-coded according to the tissue origin. **d** Pie chart showing the number of cells and proportion of cells from each tissue after filtering low-quality cells and doublets. **e** t-SNE visualization of all annotated major cell types from the 20 tissues. Cells are color-coded according to cell types. **f** Bar graph and intersect plots showing the presence of the 58 cell types across the 20 tissues. Source data are provided as a Source Data file. Schematic diagrams in **a** and **b** were created with BioRender.com.

cell types with significantly enriched markers (Fig. 1e, Supplementary Fig. S2, Supplementary Data 4). The cell types in each organ were visualized by using t-SNE, which revealed a great diversity of cell types within each tissue ranging from six cell types (subfornical organ) to sixteen cell types (lung) per tissue/organ (Fig. 1f, Supplementary Fig. S2, Supplementary Data 3). With the transcriptional profiles from these large amounts of cells, we comprehensively characterized the cell types in all tissues. For example, beige adipocytes (*CIDEA+*, *TBX1+*) were only identified in adipose tissues. Cardiomyocytes (*ACTN2+*, *MYH7+*, *FHL2+*, and *TNNT2*) were obtained from the heart. Enterocytes (*CHP2+*, *FABP1+*, and *FABP6+*) are only detected in the intestine. Cholangiocytes (*MMP7+*, *SPP1+*, and *ONE-CUT1+*), hepatic stellate cells (*ACAT2+*, *COL1A1+*, *RBP1+*, and *COLEC11+*), hepatocytes (*GHR+*, *HAMP+*, *HSD11B1+*, and *RPP25L+*), and Kupffer cells (*CD163+* and *VSIG4+*) were identified in the liver. In the lung, the AT1 (*AGER+*, *AQP5+*, *CLIC5+*, and *SCGB1A1+*) and AT2 (*SFTPB+*, *SFTPD+*, and *ABCA3+*) were identified. Immune cell types such as T cells, B cells, and macrophages are the major cell types found in PBMC and the spleen. Additionally, several types of neuron cells and microglia were identified in brain regions (Fig. 1f, Supplementary Data 3). Moreover, several cell types commonly shared between tissues were identified, such as ECs across nineteen different tissues (except PBMC); microglia mainly across all brain regions and retina; immune cells such as T cells, B cells, and NK cells across multiple different tissues (Fig. 1f). These results demonstrated that the main tissue-specific cell types were identified in each tissue and some cell types were distributed across tissues. It is consistent with the understanding that the cells derived from the three germ layers are widely distributed within the human body[34]. To further facilitate the sharing and utility of the resource generated by this study, we constructed a single-cell transcriptome atlas database (https://dreamapp.biomed.au.dk/pigatlas/).

**Validation of intra-tissue cell heterogeneity in the retina and kidney**. To further explore and validate the intra-tissue cell heterogeneity, we selected two tissues, retina and kidney, which have been largely used as models for ophthalmology and kidney diseases[35–39].

The porcine retina contains several retina-specific cell types, such as bipolar cells, cone photoreceptor cells, Müller glia, retinal ganglion cells, and rod photoreceptor cells (Fig. 2a). The main cell types shared in multiple tissues are microglia and T cells. Bipolar cells express high levels of *TRPM1*, *PCP2*, and *GNG13*. Cone photoreceptor cells exhibit a high level of *ARR3*. Microglia were indicated by the specific expression of *C1QA*, *C1QB*, *CSF1R*, and *CD68*. Müller glia demonstrated a specific expression of *RLBP1* and *CA2*, and retinal ganglion cells were annotated with high expression of *NEFL*, *THY1*, and *NRN1*. Rod photoreceptor cells had a high expression of *PDE6A*, *CNGA1*, and *SAG* (Fig. 2b, Supplementary Data 3, 4). Gene ontology (GO) analysis further showed that axonogenesis, axonal/axonal-dendritic transport, negative regulation of neurogenesis, regulation of neuron projection development, and

negative regulation of neuron differentiation are enriched in bipolar cells. In addition, pathways such as axonogenesis, gliogenesis, and negative regulation of neurogenesis, regulation of neuron projection development are enriched in Müller glia. Retinal ganglion cells share many enriched pathways with Müller glia. One specific pathway enriched in retinal ganglion cells (RGC) is neuron recognition, which is in good agreement with RGC's function in pattern recognition and visual processing[40,41]. For rod photoreceptor cells, pathways such as photoreceptor cell differentiation, photoreceptor cell cilium, photoreceptor outer segment, detection of light stimulus, and phototransduction are enriched (Fig. 2c, Supplementary Data 5). Subsequently, we validated the canonical cell markers in the retina cell types by protein staining. Rhodopsin RHO, which is essential for vision[42], was highly expressed in the rod photoreceptor cells. Arrestin 3 (ARR3), which is a non-visual arrestin and binds phosphorylated G protein-coupled receptors, was highly expressed in the cone photoreceptor cells. The G Protein Subunit Gamma 13 (GNG13) was detected strongly expressed in bipolar cells, in line with the previous findings by scRNA-seq of retina bipolar neurons[43]. CRX is a cone-rod homeobox gene expressed in cone-rod photoreceptor cells. Lastly, CDHR1 and RBP3 were identified in the photoreceptor cells (Fig. 2d, e).

In the pig kidney, collecting duct cells were characterized by the specific expression of *AQP3*, *GATA2*, and *AQP2*. Distal convoluted tubule cells were indicated by the specific expression of *TMEM52B*. Proximal tubule cells were demonstrated with the specific expression of *CUBN*, *LRP2*, *SLC13A3*, and *SLC34A1*. In addition, we identified podocytes (*NPHS1+*, *NPHS2+*, *WT1+*, and *CLIC5+*) and Loop of Henle cells (*SLC12A1+*). In addition to these cells, we identified a few other cell types in the kidney. Briefly, fibroblasts had a high expression of *CALD1*, *COL6A1*, and *DCN1*, and the T cells demonstrated enrichment of *CD3E*, *CD3D*, *CD2*, and *CD3G*. The ECs showed an increased expression of *PECAM1* and *NRP1* (Fig. 2f, g, Supplementary Data 6). We next performed GO enrichment analysis of the selected cell types. Our results show that pathways i.e., metal ion, sodium ion transmembrane transport, cellular drug response were mainly enriched in collecting duct cells. The positive regulation of sodium ion transmembrane transport, regulation of transmembrane transport, and regulation of membrane potential were enriched in distal convoluted tubule cells. In addition, the cell-matrix adhesion, regulation of angiogenesis, ficolin-1-rich granule lumen, and myofibril were enriched in collecting duct cells, ECs, and podocytes. Glomerular epithelial cell differentiation, renal filtration cell differentiation, nephron development, and glomerulus development were specifically enriched in podocytes. Moreover, our single-cell analysis suggested that loop of Henle cells plays an important role in potassium ion homeostasis, potassium ion import, chloride ion homeostasis, and metanephric nephron tubule development (Fig. 2h, Supplementary Data 6), in line with its functions in maintaining iron and water homeostasis[44]. To validate the canonical markers expressed in the kidney, we further analyzed them with immunohistochemistry. Our results showed that NPHS2, a podocyte-specific marker, is specifically expressed in podocytes. SLC12A1 was specifically expressed in the loop of Henle cells, while GATA2,

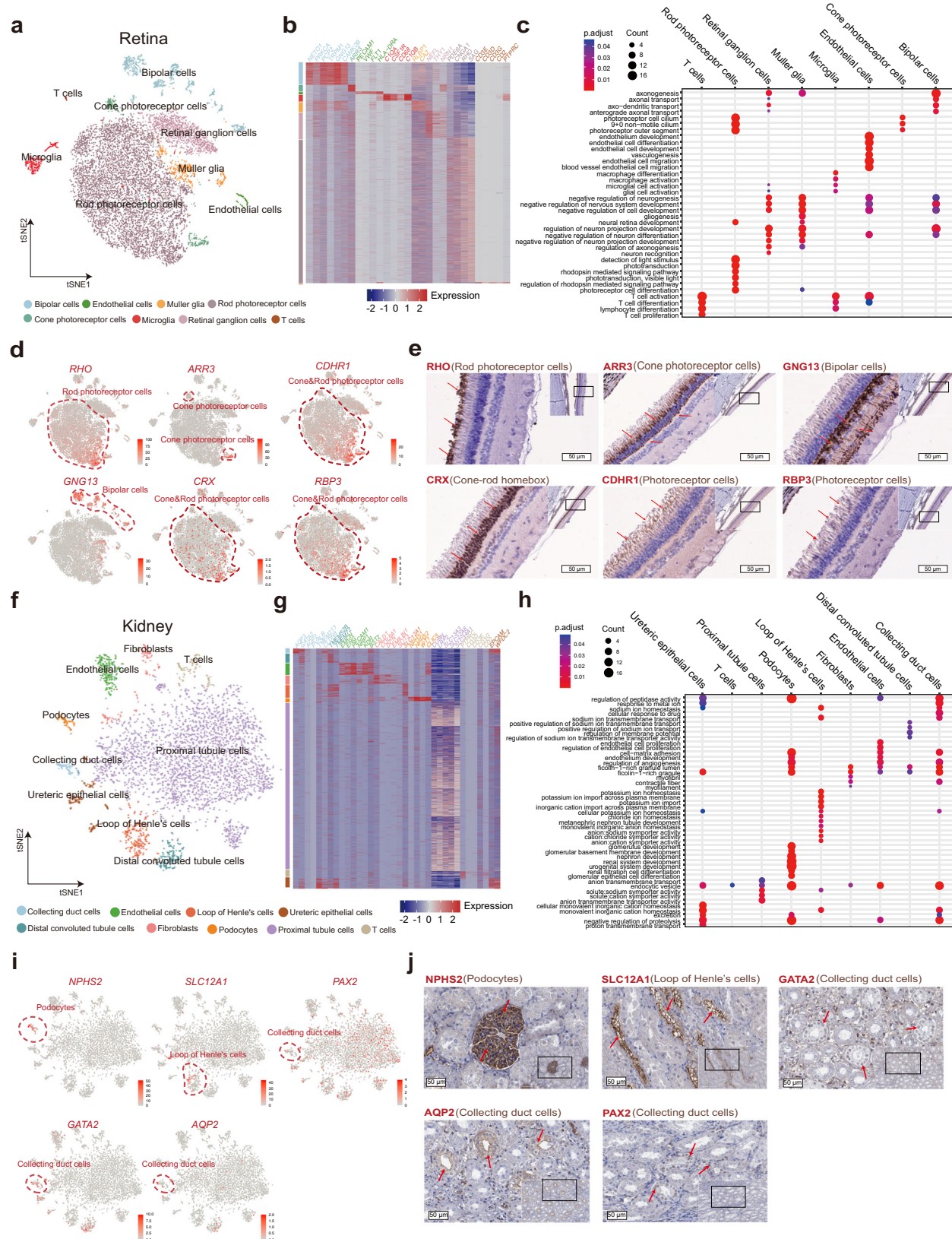

AQP2, and PAX2 were specifically expressed in collecting duct cells in the porcine kidney (Fig. 2i, j).

**Endothelial cell heterogeneity**. Single-cell transcriptome analysis also enables us to probe rare cell types. Endothelial cells (ECs), which line the vascular systems and play important roles in e.g. regulating immune responses, regulation of blood fluidity, cardiovascular homeostasis, maintenance of vascular functions, have been extensively studied by scRNA-seq in human and mouse[45–49], though little is known about the single-cell transcriptome and heterogeneity of ECs in pigs. We therefore focused

**Fig. 2 Validation of cell heterogeneity in retina and kidney. a** t-SNE visualization of major cell types in the retina. **b** Heatmap of marker gene expression in the cell types captured in the retina. **c** GO term enrichment analysis on marker genes in the major cell types in the retina. The hypergeometric test was used for GO term analysis, and *p* values were adjusted by Benjamini & Hochberg. **d** t-SNE visualization of the expression of *RHO, ARR3, CDHR1, GNG13, CRX,* and *RBP3* in the retina. **e** Representative IHC staining of RHO, ARR3, GNG13, CRX, CDHR1, and RBP3 in pig retina (*n* = 3). **f** t-SNE visualization of major cell types in the kidney. **g** Heatmap of marker gene expression in major cell types in the kidney. **h** GO term enrichment analysis on marker genes of major cell types in the kidney. The hypergeometric test was used for GO term analysis, and *p* values were adjusted by Benjamini & Hochberg. **i** t-SNE visualization of *NPHS2, SLC12A1, PAX2, GATA2,* and *AQP2* expression in the kidney. **j** Representative IHC staining of NPHS2, SLC12A1, GATA2, AQP2, and PAX2 in pig kidney (*n* = 3). Source data are provided as a Source Data file.

on the ECs which were captured in 19 tissues in our datasets. To characterize the heterogeneity of ECs in pig organs, we extracted ECs from all 19 tissues expressing the canonical EC marker *PECAM1*, and excluded epithelial cells (*EPCAM*+), immune cells (*PTPRC*+), fibroblasts (*COL1A1*+), and pericytes (*PDGFRB*+) which frequently co-express *PECAM1* (Fig. 3a, Supplementary Fig. S3a, Supplementary Data 7). In total, we obtained 9520 ECs from 19 tissues and 56% of all ECs were derived from the adipose tissues (Supplementary Data 7). This is expected, as we applied a protocol optimized for enriching ECs from adipose tissues (see methods). After batch correction, all ECs were separated into 21 subtypes with significantly enriched genes (Fig. 3b, c, Supplementary Fig. S3b, Supplementary Data 8). Although the number of analyzed ECs from each tissue are largely different, the distribution of these 21 EC clusters was quite similar between the tissues (Fig. 3c). We performed gene ontology analysis of these 21 EC clusters based on enriched markers (Supplementary Data 8). Two EC clusters with distinct functions were highlighted here. The EC cluster (c12) upregulates defense marker genes (i.e., complement: *C1QA, C1QB, C1QC*; cathepsins: *CTSS, CTSD, CTSB, CTSZ*; cystatins: *CST3, CSTB* (Supplementary Fig. S3c, Supplementary Data 8)), suggesting that it is an EC phenotype with immune active features. This observation is consistent the scavenging and immune-modulating EC phenotype reported previously[50–53]. In adult tissues, most ECs are quiescent. We identified a small EC cluster (c17) significantly up-regulating cell proliferation genes (*CENPF, CENPE, TOP2A, TPX2*) (Supplementary Data 8), which we define as proliferating ECs. The presence of proliferating ECs in the analyzed pig tissues agrees with the age of the pigs used for this study (Supplementary Data 1).

As EC heterogeneity can be caused by tissue types and vascular bed, we therefore focused on the ECs from adipose tissues and further investigated the EC heterogeneity within tissues. We next performed cell clustering based on the adipose tissue-derived ECs and annotated each ECs subtype based on the expression of ECs markers and pseudotime trajectory analysis. The ECs from adipose tissues are mainly composed of blood ECs (arterial, capillary, and vein) and lymphatic ECs (Fig. 3d, e), which also seems to express higher level of mesenchymal genes (Supplementary Fig. S3a). We also identified three functionally distinct phenotypes: proliferating EC, immune active EC and an EC phenotype with co-expression of the mesenchymal cell markers *ACTA2* and *TAGLN* (Fig. 3d, Supplementary Data 9). Both immune active and proliferating ECs have been described above and in previous scRNA-seq studies of mouse and human ECs[45–47]. Here we focus on the small fraction of mesenchymal-like endothelial phenotype. Previous studies have suggested the existence of such a specific endothelial phenotype, which is undergoing the processing of endothelial-to-mesenchymal transition (EndMT)[54–57]. The number of expressed genes per cell is similar between EndMT cells and other EC phenotypes (Supplementary Fig. S3d), confirming that it is a cluster of transcriptomically distinct EC subtypes which is not caused by doublets of an EC and a mesenchymal cell. To further validate the mesenchymal-like EC phenotype, we performed pseudotime

trajectory analysis which demonstrates that the EndMT cells undergo a dynamic developmental transition from ECs to mesenchymal cells (Fig. 3f). During the EndMT process, expression of EndMT inducing genes (*TGFB2* and *SNAI1*) were high in the early EndMT process, followed by gradually decreasing expression of EC-specific genes (e.g., *PECAM1, VWF, ICAM1,* and *CDH5*), and the increased expression of mesenchymal cell-specific genes (e.g., *ACTA2, TAGLN, CD44, VIM,* and *CNN1*) (Fig. 3g, h). Early findings suggested that TGF-β2 is a key regulator of the EndMT process[55,58–61]. Our pseudotime analysis of EndMT cells also suggested that the expression of the TGF-β signaling pathway (*TGFB2*) a key inducing factor driving the EndMT process (Fig. 3h).

To verify some of the markers for EndMT cells at the protein and histological level, we analyzed the co-expression of ECs markers (*PECAM1* and *VWF*) and mesenchymal cell markers (*ACTA2* and *TAGLN*) by antibody-based immunofluorescence staining. Our results showed that EndMT cells can be clearly identified in adipose tissues with co-expression of *VWF* and *TAGLN* (Fig. 3i), as well as *PECAM1* and *ACTA2* (Fig. 3j), in a small fraction of ECs. Corroborating the scRNA-seq results, only a very small fraction of the adipose tissue ECs is EndMT. We also validated another ECs-specific marker *FABP4* found by our single-cell analysis of pig adipose ECs. The fatty acid-binding protein 4 (FABP4) is a lipid transport protein, which is expressed in adipocytes and capillary ECs[62]. To investigate if *FABP4* is expressed in capillary ECs across pig tissues, we analyzed pig liver, heart, kidney, spleen, duodenum, jejunum, cerebellum, and brain cortex by immunohistochemistry. The results show that FABP4 is strongly expressed in capillary ECs of all included pig tissues (Supplementary Fig. S3e) and is highly expressed in the subcutaneous adipocytes and the adipose ECs (Fig. 3k). Collectively, our results support the previous model of EndMT (Fig. 3l), which is involved in important mesenchyme-associated physiological and pathological processes[54].

**Validation of EndMT in cultured ECs.** Cultured ECs are common models for studying the EndMT process, of which differentiation of ECs into mesenchymal cells can occur spontaneously or through TGF-β induction[63]. Previously, we also identified the EndMT ECs phenotype in cultured human lung tumor ECs[46]. To further validate this EndMT ECs phenotype, we isolated and cultured primary ECs from pig lung and aorta and analyzed them by scRNA-seq (Fig. 4a). In total, after filtering low-quality cells and doublets, 5698 and 882 cells were obtained from the cultured ECs of pig lung and aorta respectively, which were further clustered into 5 clusters (Fig. 4b–f) based on the expression of cell-type-specific markers (Fig. 4c–e) and cell cycle analysis (Fig. 4f). In cultured ECs from both pig lung and aorta, we identified three clusters of proliferating ECs (G1, S and G2M phase) and an intermediate EC phenotype. Because the ECs isolation protocol is based on enzymatic perfusion of the blood vessels (see methods), we also identified a cluster of fibroblasts expressing high levels of *COL1A1* and *COL1A2* (Fig. 4b–e). In the cultured aorta ECs, we identified a cluster of cells that highly express mesenchymal cell

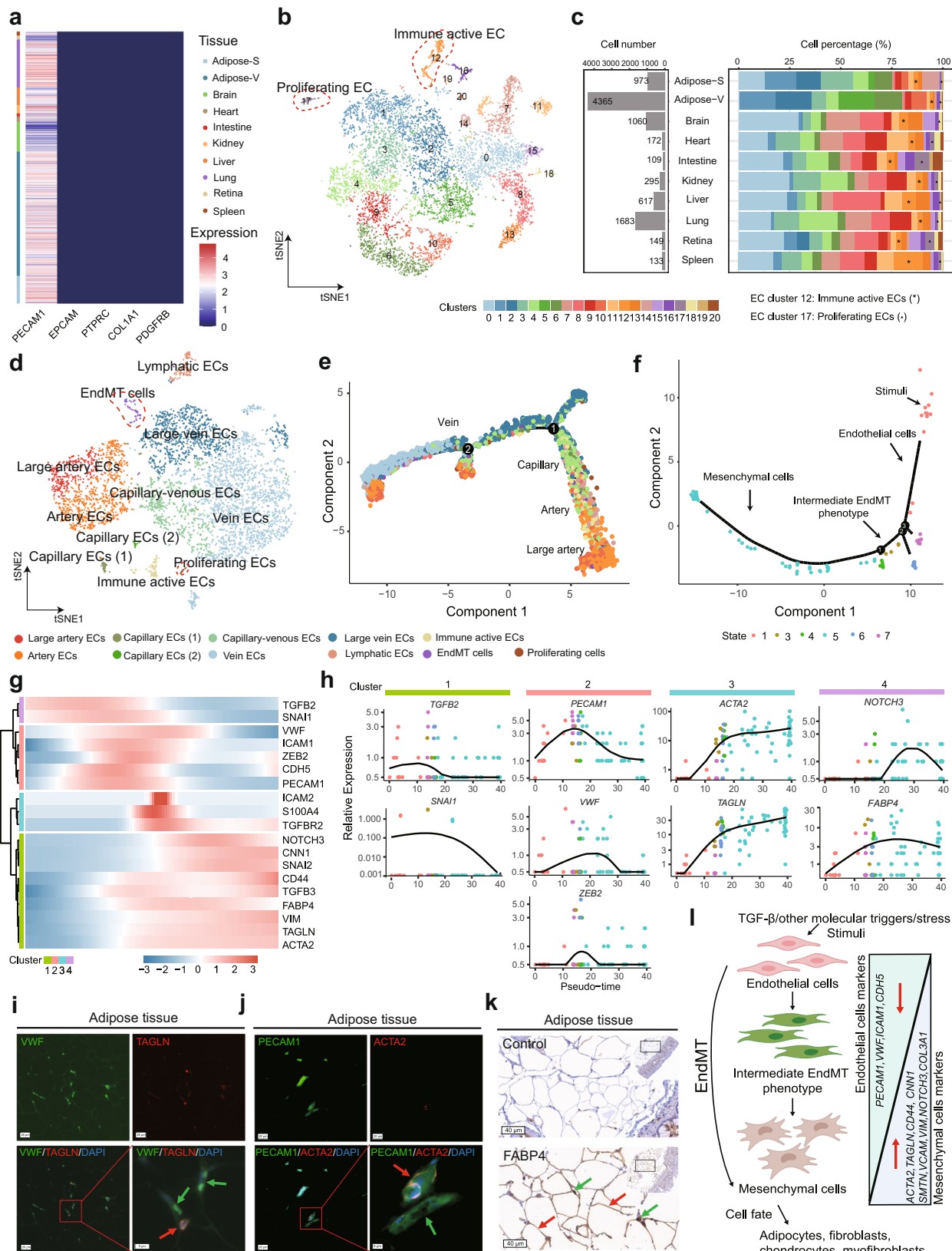

markers (*TAGLN* and *ACTA2*) but not fibroblast markers (*COL1A1* and *COL1A2*), which we defined as EndMT-like cells (Fig. 4d, e). The cultured lung and aorta ECs used for scRNA-seq were maintained in normal ECs growth medium. This EndMT-like cells might be induced by the presence of TGF-β2 in serum or contaminating mesenchymal cells from the isolation procedure.

To further validate that EndMT can be induced by TGF-β2, we treated cultured pig aorta endothelial cells (PAECs) with TGF-β2 (2 ng/mL) and measured the expression mesenchymal cell marker ACTA2 and ECs marker CD31 by fluorescence-conjugated antibody staining and flow cytometry analysis (Supplementary Fig. S4a). Increased ACTA2 expression was already detected in

**Fig. 3 Single-cell transcriptome analysis of porcine ECs in pig tissues. a** Heatmap of gene expression in the selected *PECAM+/EPCAM−/PTPRC−/ COL1A1−/PDGFRB−* ECs from 19 tissues. **b** t-SNE visualization of 21 EC clusters, which are colored according to EC cluster numbers. **c** Bar graph shows numbers and percentage of ECs in tissues. ECs were colored according to clusters. **d** EC subtypes in adipose tissues. Cells are color-coded according to the subtypes of ECs. **e** EC subtypes trajectory analysis in adipose tissues (adipose-S and adipose-V) using monocle 2 and cells on the tree are colored by EC subtypes. **f** EndMT EC trajectory analysis using monocle 2 and cells on the tree are colored by EndMT states. **g** Pseudotime trajectory analysis of marker genes in different states of EndMT cells. **h** Representative pseudotime trajectory of marker genes (*TGFB2, SNAI1, PECAM1, VWF, ZEB2, ACTA2, TAGLN, NOTCH3, FABP4*) in different states of EndMT ECs. **i** Representative Immunofluorescence staining images of VWF and TAGLN in adipose tissues. Arrow (red) indicates ECs expressing both VWF and TAGLN. Arrow (green) indicates ECs only expressing VWF ($n = 3$). **j** Representative Immunofluorescence staining images of PECAM1 and ACTA2 in adipose tissue. Arrow (red) indicates ECs expressing both PECAM1 and ACTA2. Arrow (green) indicates ECs only expressing PECAM1 ($n = 3$). **k** IHC of FABP4 in ECs and adipose tissues. Arrows (green) indicate ECs. Arrows (red) indicate adipocytes. Control was stained with an antibody against a gene not expressed in adipose tissues ($n = 3$). **l** An integrated model of EndMT process. Source data are provided as a Source Data file. Schematic diagrams in **l** were created with BioRender.com.

cultured PAECs 2 days after TGF-β2 treatment (Supplementary Fig. S4). After culturing PAECs in medium with TGF-β2 for five days, the expression of ACTA2 was significantly increased in PAECs by 6 folds compared to untreated controls (Fig. 4g, $P = 1.44\text{E-}7$, *t*-test). The CD31 expression was slightly decreased in the TGF-β2 treated PAECs, but not significant ($P = 0.300$, *t*-test). Our results also showed that TGF-β2 significantly inhibits the proliferation of ECs (Supplementary Fig. S4b, $P = 1.33\text{E-}4$, *t*-test). Consistent with the cultured PAECs, similar TGF-β2 induced EndMT results were also obtained in cultured human umbilical vein endothelial cells (HUVECs) (Fig. 4h). Together, our results demonstrated that EndMT is driven by the TGF-β2 signaling pathway and is conversed between pig and human.

**Communications between ECs and other cell types.** Intercellular communication between ECs and other parenchymal cells in the tissue plays a vital role in the structure and function of maintaining normal tissue growth, development, and homeostasis[64]. We focused on the ECs and analyzed the communications between ECs and other cell types using CellChat, a tool for quantitatively inferring and analyzing intercellular communications based on the differential expression of ligand and receptor gene pairs[65]. For this analysis, we used single-cell transcriptome data from the liver, kidney, and heart, which are the three most studied pig organs in biomedical research[66]. We performed a comparison with corresponding scRNA-seq datasets of the human liver, kidney, and heart (see methods) to gain a better understanding of the cell–cell interactions in these three organs between pig and human. We identified six cell types in the liver (hepatocytes, ECs, Kupffer cells, B cells, T/NK cells, and erythroid cells), six cell types in the kidney (epithelial cells, podocytes, proximal tubule cells, collecting duct cells, ECs, and distal convoluted tubule cells), and five cell types in the heart (ECs, fibroblasts, cardiomyocytes, lymphoid cells, and myeloid cells) isolated from both pig and human tissues (Supplementary Data 10, 11).

We next investigated signaling interactions based on ligand-receptor pairs and performed a global communication pattern recognition analysis to identify the key signal communications involved in ECs. Our results suggested that ECs use the vascular endothelial growth factor (VEGF)[67], PDGF[68], TGF-β[69], and BMP[70] signaling pathways as major communicating pathways with other cell types in the liver, kidney, and heart in both human and pig. However, single-cell-based cell communication analysis provided several unique insights regarding tissue-specific and cell-type-specific communications between ECs and other cell types. The VEGF signaling is the most studied pathway in ECs activation, proliferation, and angiogenesis. In the liver and heart, our results showed that there are similar sender-receiver communications between ECs and other cell types. In the kidney, the VEGF signaling pathway is used by most renal cells for cell–cell communications. Particularly, similar between pigs and humans, there is a higher level of communication between ECs,

podocytes, and proximal tubule cells through the VEGF pathway (Fig. 5a, Supplementary Fig. S5a). The analysis suggests that the VEGF signaling pathway can be used by most renal cells for intercellular communication. Unlike the VEGF signaling pathway, ECs only communicate with a few other cell types through the PDGF pathway. For example, pig ECs communicate with immune cells (Kupffer-, T-, NK-, and B cells) in the liver, with all cell types in the heart, and with only podocytes in the kidney. Human ECs communicate with hepatocytes and Kupffer cells in the liver, with fibroblasts in the heart, and with podocytes and proximal tubule cells in the kidney through the PDGF pathway (Fig. 5b, Supplementary Fig. S5a). Similarly, analysis of TGF-β and BMP pathways-mediated cell–cell communications show similar cell-type-specific and tissue-specific preferences, as well as some divergence between pig and human (Supplementary Fig. S5b). The presence of TGF-β-based cell–cell communication between ECs and other tissue types further suggests a potential cellular mechanism in inducing EndMT in tissues. However, it should be noted that the inferred cell–cell communications based on single-cell RNA sequencing could be affected by e.g., posttranscription or posttranslational protein modifications, completeness and biases in the genome annotation between pig and human. We demonstrated that the pig single-cell RNA atlas provides a valuable resource for inferring cell–cell communications within pig tissues or between pig and human tissue.

**MEF2C is a conserved regulon in microglia evolution.** Single-cell transcriptomic analysis not only provides good insights into the cellular heterogeneity and functional diversity of structural cell types across tissues, but it is a good way to uncover the similarities and divergences of cell types across species. In this study, we utilized our single-cell pig atlas to analyze the cross-tissue ECs heterogeneity and ECs conversion in adipose tissues. We were also interested in the cross-species cell types, which are mainly focused on microglia in the brain across 13 species. It enabled us to better understand cell-type evolutions. The pig single-cell transcriptome atlas includes nine brain regions (Fig. 6a). Most cell types in the brain were clustered based on the regions (Fig. 6b, Supplementary Fig. S6a). We also validated a few cell types by immunohistochemistry, confirming that scRNA-seq is a robust method for the classification of different cell types (Fig. 6c, Supplementary Fig. S6b), including SLC1A6 which is a marker for Purkinje cells[71] expressed in pig cerebellum; CALB2 which is an inhibitory interneuron marker; PAX6 which is a highly specific marker for granule cells in cerebellum. By protein staining, we also detected SLC17A7 in the excitatory synapse in the pig hippocampus. Lastly, protein staining of SST is consistent with scRNA-seq suggesting that SST is a specific marker for inhibitory interneurons in the pig cortex (Fig. 6c).

We focused on microglia which is important for brain homeostasis and involved in several brain disorders i.e.

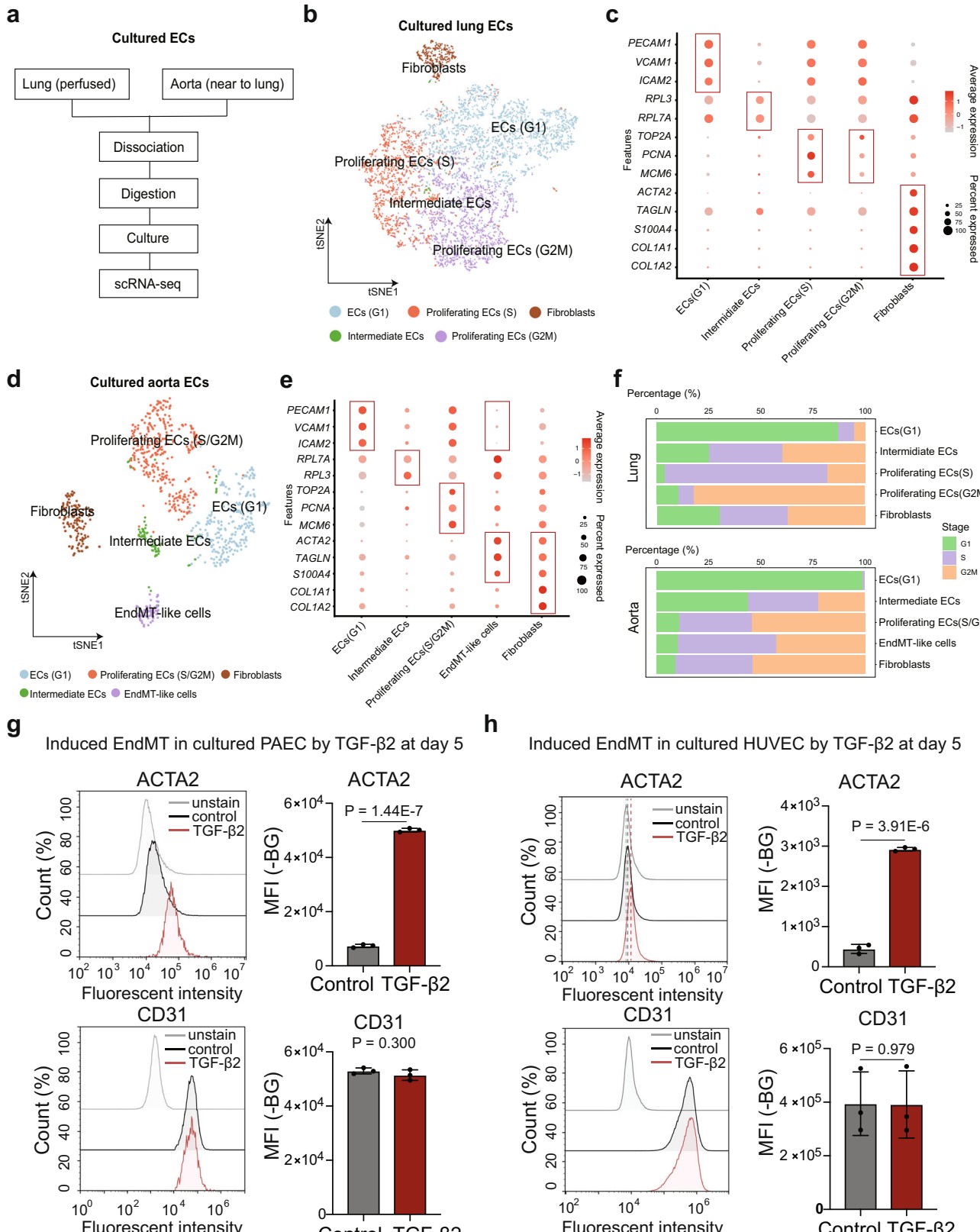

**Fig. 4 Validation of EndMT ECs in cultured ECs from pig lung and aorta by scRNA-seq. a** The flowchart of ECs isolation and culture from pig lung and aorta. **b** tSNE visualization of cell-type annotations from cultured lung ECs. **c** Dot-plot of selected canonical marker genes for cell-type annotations from cultured lung ECs. **d** tSNE visualization of cell-type annotations from cultured aorta ECs **e** Dot-plot of selected canonical marker genes for cell-type annotations from cultured aorta ECs. **f** Cell cycle analysis of annotated cell types in cultured lung and aorta ECs. **g** Expression of ACTA2 and CD31 in cultured PAEC treated with TGFb2 for five days ($n = 3$, two-sided $t$-test). Values are presented as mean±SD. **h** Expression of ACTA2 and CD31 in cultured HUVEC treated with TGFb2 for 5 days ($n = 3$, two-sided $t$ test). Values are presented as mean ± SD. Source data are provided as a Source Data file.

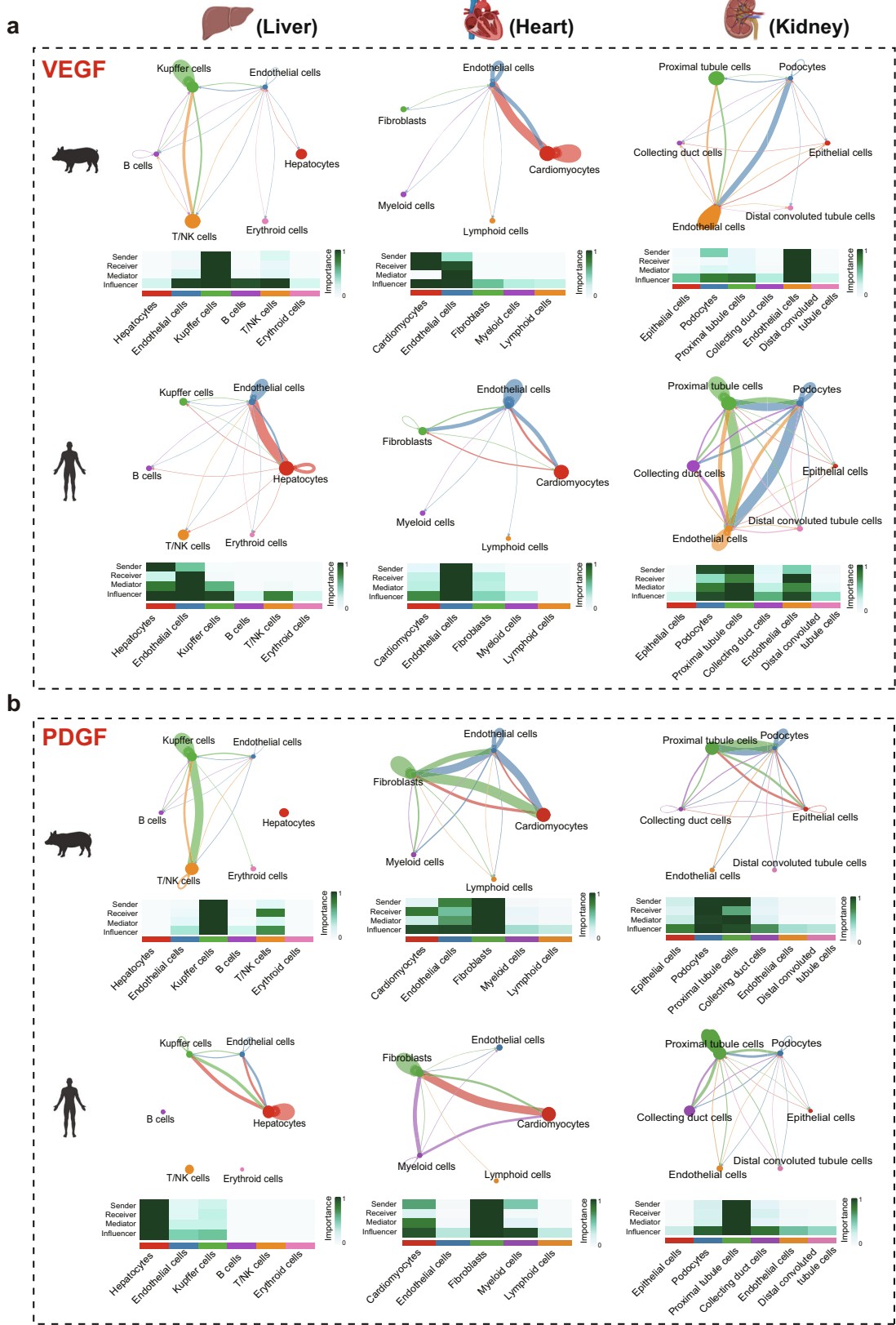

**Fig. 5 Comparison of cell communication and signaling pathway between pig and human. a** Comparison of cell–cell communication of VEGF signaling pathway in liver, heart, and kidney. **b** Comparison of cell–cell communication of PDGF signaling pathway in liver, heart, and kidney. Source data are provided as a Source Data file.

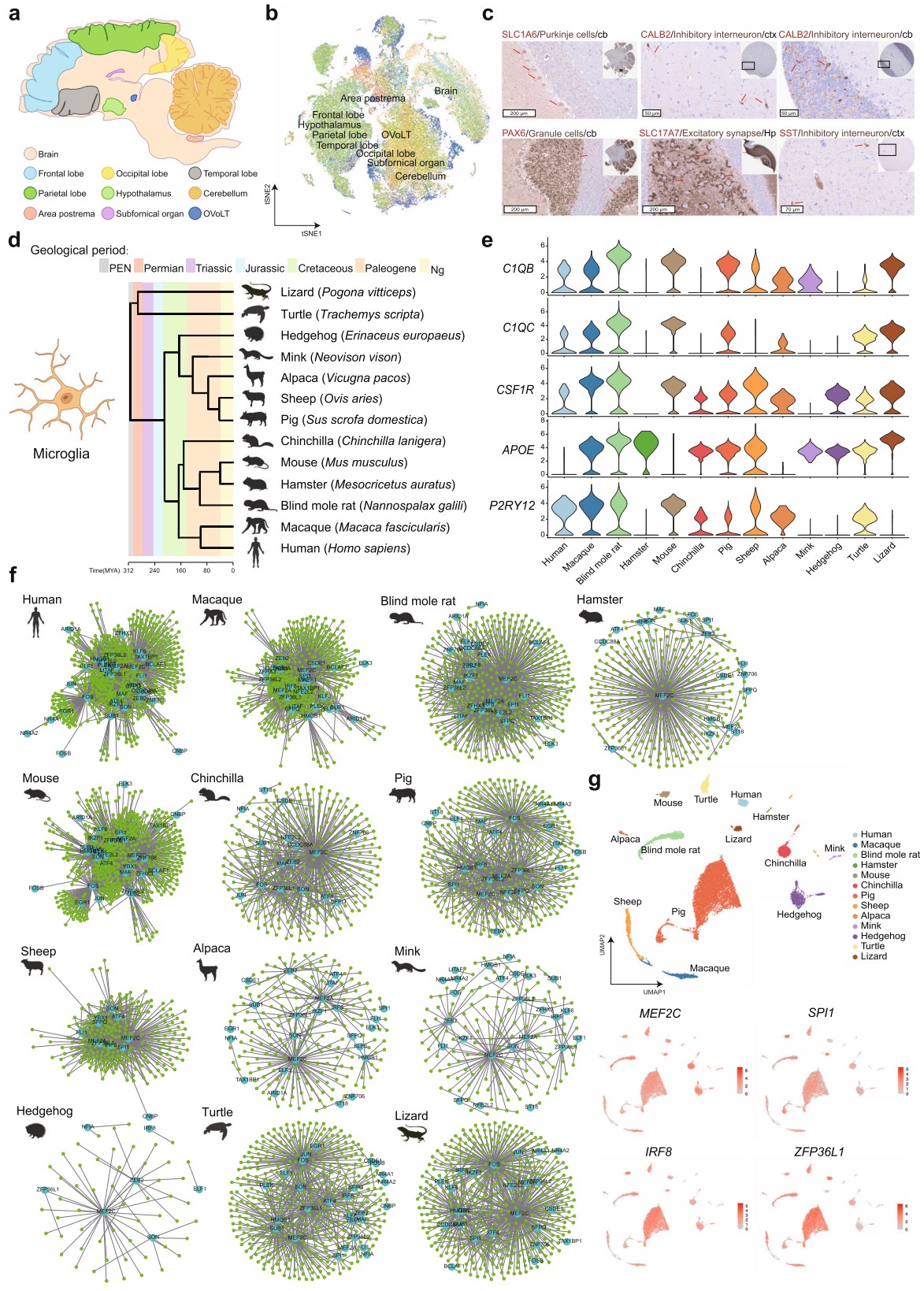

amyotrophic lateral sclerosis[72]. Microglia are the primary resident immune cells of the brain, which play a critical role in many physiological and pathological brain processes. The conserved and divergent microglia gene program in pan-species provides important implications for investigating the microglia evolutionary module in human brain diseases[73]. To characterize the conservation and divergence of microglia in pigs, we analyzed the single-cell microglia datasets from 13 different species spanning more than 300 million years of evolution. The species contain bearded dragon lizard, turtle, hedgehog, mink, alpaca,

**Fig. 6 Validation of brain cell types and comparison of microglia regulome across species. a** Schematic diagram visualizing the 9 different brain regions analyzed by this study. **b** t-SNE visualization of cells in the 9 brain regions. Cells were color-coded according to brain regions. Cell types were shown in extended Fig. S6. The hypergeometric test was used for GO term analysis, and p values were adjusted by Benjamini & Hochberg. **c** Representative IHC of SLC1A6, CALB2, PAX6, SLC17A7, and SST by antibody staining in the cortex or cerebellum. Arrows indicate corresponding cell types (n = 3). **d** Phylogenetic tree based on the NCBI taxonomy of animals used in this study (generated via http://www.timetree.org//). **e** Violin plots visualizing the expression of canonical microglia markers in the 13 species. **f** Conserved genetic regulatory networks in microglia within each indicated species. Light blue nodes represent regulators and green nodes represent corresponding target genes. **g** Feature plots visualizing the clustering of microglia single-cell transcriptome and expression of four TFs among the 13 species. Source data are provided as a Source Data file. Species icons in **d**, **f** were created with BioRender.com.

sheep, pig, chinchilla, mouse, hamster, blind mole rat, macaque, and human (Fig. 6d, Supplementary Data 12). The microglia-specific marker genes *C1QB* and *C1QC* were expressed in most species except for Chinchilla, Hamster, and Hedgehog, while the markers of *CSF1R*, *APOE*, and *P2RY12* are covertly expressed in all 13 species (Fig. 6e), suggesting conservation and divergence in the expression of microglia markers across species.

Transcription factors (TFs) have been demonstrated as the important regulators of gene expression and with the ability to shape different phenotypes of microglia[74]. We, therefore, performed single-cell genetic regulatory network (GRNs) inferring and clustering analysis to assess TFs underlying differential gene expression in microglia across species. In total, 1590 conserved TF-target pairs were identified, which were observed in at least five of thirteen species. Interestingly, two pairs of TF-target were conserved in ten different species, which are MEF2C_P2RY12 and MEF2C_ZFP36L1. The target *P2RY12* is a microglia-specific gene, while the target *ZFP36L1* is reported as a pivotal regulator for microglial fate specification[75,76]. For TF_target pairs covering nine species, four pairs of conserved TF_target were identified including MEF2C_C1QB, MEF2C_FAU, MEF2C_RGS10, and MEF2C_SERBP1. The target *C1QB* is a microglia-specific marker. The target *RGS10* is reported as a key regulator of proinflammatory cytokine produced in microglia for neuroprotective factors[77]. *SERBP1* plays an important role in proinflammatory TLR4 signaling[78]. However, the *FAU* gene encodes a ubiquitin-like protein fused to the ribosomal protein S30. These results demonstrated that MEF2C is a core TF regulating multiple key genes related to microglia functions. Furthermore, the TF-target pairs in eight species share the unique TF of MEF2C, and most of the conserved TFs in at least five species also mainly contain MEF2C. These targets of MEF2C include most microglia and immune-related genes such as *PTPRC*, *CSFER1*, *APOE*, *AIF1*, *CD14*, and *CTSS* et. These results demonstrated that MEF2C is a conserved TF in microglia evolution. It is consistent with the functional TF of MEF2C reported regulating the fundamental functions of microglia[79]. The other importantly conserved TFs were identified for microglia functions such as *SPI1* and *IRF8* (Supplementary Data 13).

Subsequently, we separately analyzed the conserved TF-target pairs shared by five of thirteen different species. For example, in humans, the top ten of these conserved TFs were shared with four other random species are *ARID1*, *ATF4*, *BCLAF1*, *CCDC88A*, *CNBP*, *CSDE1*, *EGR1*, *ELF1*, *FLI1*, and *FOS*. In the macaque, the top ten conserved TFs are *ELF1*, *FLI1*, *IKZF1*, *IRF8*, *KLF6*, *LITAF*, *MEF2A*, *MEF2C*, *NFE2L2*, and *SFPQ*. In pigs, the top ten conserved TFs are *MEF2C*, *IRF8*, *MEF2A*, *SON*, *SPI1*, *ATF4*, *ELF1*, *FLI1*, *FOS*, and *HMGB1*. However, for reptiles such as lizards, the top ten conserved TFs are *MEF2C*, *ATF4*, *IRF8*, *MEF2A*, *MEF2C*, *ZEB2*, *SPI1*, *ATF4*, *FOS*, and *HMGB1*. These conserved TFs in each species contain the shared conserved TF-target pairs and species-specific TF-target pairs (Fig. 6f, Supplementary Data 13). Additionally, the rank top ten conserved TFs (*MEF2C*, *ZFP36L1*, *FOS*, *MEF2A*, *IRF8*, *SON*, *HMGB1*, *ZEB2*, *SPI1*, and *ATF4*) in humans were indicated with the enrichment

of pathways such as adaptive immune response, regulation, or humoral immune response in the immune system, and the central nervous system neuron differentiation and development for neuron-related pathways (Supplementary Fig. S6c). Collectively, these results indicate that the combinations of multiple TFs regulate microglia development and maintain the functional states of microglia. Furthermore, we investigated the expression level of conserved TFs of *MEF2C*, *SPI1*, *IRF8*, and *ZFP36L1* across species. The results showed these conserved TFs are highly expressed in microglia in these 13 species (Fig. 6g). Collectively, we provide a valuable resource of the conserved and divergent GRNs program for microglia evolution across species, with important implications for future development of the microglia functional studies in the brain.

## Discussion

Pigs are important large animal models for studying complex human diseases, as well as promising alternative organ donors for humans due to their high similarities with humans: physiology, anatomy, genetics, metabolism, and organ size[4]. Single-cell transcriptomic profiles of multiple organs in the mouse and human body reveals the cellular compositions and heterogeneity of inter-and intra- organs and offers the opportunity to overall organ development, physiology, and plasticity[22,34,80,81]. This study generated a multiple-organ single-cell transcriptomic atlas of pigs covering twenty tissues. The landscape profiles of pig organs depicted the transcriptomic cellular heterogeneity in each tissue and expand the functions of cross-tissue cell types and rare cell-type identification. With a focus on endothelial cells, we identified the subpopulations of cross-tissue ECs, such as blood ECs, lymphatic ECs, and several subtypes of functionally distinct ECs. These cell types were also reported in the human single-cell atlas. We are also able to identify functionally specific ECs, such as the immune active ECs that express typical ECs markers (*PECAM1*+) and immunomodulating genes (*CD68*+, *C1QA*+, *C1QB*+, and *AIF1*+). These findings of immune active ECs are in good agreement with our previous scRNA-seq based ECs taxonomy in mouse and human tissues[45–47] and the well-study immune-modulating functions of ECs[82–84]. In this study, the ECs were only enriched from adipose tissues through a modified isolation protocol. The degree of ECs heterogeneity was not revealed as detailed as the mouse ECs atlas[45]. To systematically characterize and compare the ECs heterogeneity across different tissues and organs by scRNA-seq, future studies are needed to be performed using enriched the ECs by MACS and/or FACS.

Angiogenesis is matured through ECs and non-ECs to form the vascular channels. The non-endothelial microcirculation is called vascular mimicry[85]. In physiologic and pathological angiogenesis, macrophages are thought to play a supportive role to promote vascular mimicry outgrowth through cytokine secretion and remodeling of the extracellular matrix (ECM)[86,87]. In tumors, macrophages are believed to reprogram the pathological angio-genesis to serve as the major source of angiogenic factors[88]. PECAM1 signaling participates in the regulation of leukocyte

detachment, T cell activation, and angiogenesis[89]. Therefore, the high expression of macrophage marker genes and *PECAM1+* cell population may play an important role in vascular mimicry during angiogenesis. EndMT ECs were identified in the pig adipose tissues and cultured ECs with co-expression of ECs markers of *PECAM1* and *VWF*, and the mesenchymal cell markers of *ACTA2* and *TAGLN*. In particular, the EndMT cells underwent the dynamic transition stages with the decreased expression of the ECs markers *PECAM1* and *VWF* while the expression of the mesenchymal cell markers *ACTA2* and *TAGLN* increased. This phenotype is consistent with the stages of EndMT: (1). EndMT is initiated by autocrine and/or paracrine inflammatory signals such as TGF-β2, which was also validated by our TGF-β2 induction experiment of cultured PAECs and HUVECs, or response to vascular injury; (2). Transitioning ECs acquire a migratory phenotype, invade under the vascular basement membrane, and begin to express mesenchymal markers, such as *ACTA2*; (3). Cells that have undergone EndMT have lost their endothelial phenotype. These EndMT-derived cells contribute to the local mesenchymal lineage population and are likely to produce various growth factors, such as TGF-β2[90]. EndMT is an important developmental process, participating in tumor formation, invasion, and metastasis, and has been extensively participating in several diseases by causing morphology changes and pathological processes. The mesenchymal cells derived from EndMT might be differentiated into various mesenchymal cell types for tissue engineering and subsequent transplantation into the patient[91].

The most potent pathway of VEGF played the core role in the intercellular interactions between ECs and tissue-specific cell types, suggesting active interactions and communications between ECs and other cell types in organs[92]. Angiogenesis occurs in both physiologic and pathological conditions and interplays with other cell types[93,94]. In adult tissues, most ECs are quiescent, but these ECs are metabolically active and actively involved in the regulation of several important cellular processes such as immune modulations[95]. In addition, the growth of pathological angiogenesis in human diseases such as cancers highlights that the targeting this process should help to reduce both morbidity and mortality from carcinomas[96]. Hence, anti-angiogenic therapy is a novel approach for the treatment of cancers, diabetic retinopathy, and other angiogenesis-dependent diseases[97].

Pigs are an excellent model for studying genetic and somatic evolution[23]. In this study, we demonstrated that the evolution of microglia, a cell type that exhibits increasing interest and is important in neurological disorders[98]. Approximately 1500 regulatory sequence-specific DNA-binding factors (transcription factors, TFs) are encoded in the human genome, which is upregulated in a tissue-specific and cell-type-specific manner. Changes in gene expression between species could be due to changes in the TFs and/or changes in the instructions within the regulatory regions of specific genes. TF expression patterns and binding activities could advance the understanding of how tissue specificity and conserved regulatory functions across species[99–101]. Here, we investigated the genetic regulatory networks of microglia across thirteen different species, spanning more than 300 million years of evolution. The conserved TFs across different species demonstrated the regulatory mechanism of target genes for the fundamental functions of microglia. In particular, the conserved TFs of *MEF2C* were detected in ten species, and its regulatory genes are important for microglia functions in the brain such as *P2RY12*, *ZFP36L1*, *RGS10*, and *SERBP1*. These transcriptional regulatory genes play an important role in microglia biology, such as microglia fate specification, a key regulator of proinflammatory cytokine produced in microglia, suggesting that microglia perform overall similar functions during species evolution[73].

We also highlight a few limitations of the current study. First, tissues used by this study are taken from animals of adult ages. Thus, the development and age effects on gene expression and cell-type compositions in tissues cannot be addressed by the data generated by this study. When conducting the comparison of cell-type-specific gene expression between pig and human, effects of ages, gender, and physiological conditions cannot be addressed by this study. Second, the scRNA-seq and snRNA-seq experiments were conducted by two laboratories, using two different sets of tissues, and with technologies provided by two companies. Thus, the difference in the number of cell types and fraction of each cell type captured by the two methods could be caused by the many steps during sample processing. Since the aim of this study was not focused on comparing the scRNA-seq and snRNA-seq technologies, we carried out comprehensive batch correction and normalization to use both scRNA-seq and snRNA-seq datasets to construct the first pig single-cell transcriptome atlas. Most importantly, the transcriptome of sample cell types captured by scRNA-seq and snRNA-seq is highly correlated, suggesting that both methods can capture the transcriptome of cells with high fidelity. Since the number and composition of cell types could be affected by the single-cell transcriptomics analysis methods (scRNA-seq and snRNA-seq), comparison of cell-type abundance and composition between tissues or conditions should be carried out using datasets generated with the same method to avoid method-induced biases. Lastly, we provide the first insight into EC heterogeneity in pig tissues. To fully characterize the EC heterogeneity and identify functionally distinct EC phenotypes in pig organs, an EC-focused single-cell transcriptome analysis should be carried in future study.

In summary, we constructed the first single-cell transcriptomic atlas of pig organs (https://dreamapp.biomed.au.dk/pigatlas/). We identified the tissue-specific cell types and cross-tissue cell types such as ECs, immune cells, and microglia. Moreover, some rare cell types were identified in our data, such as EndMT ECs, suggesting the rare cell types could be identified by single-cell techniques. The regulatory mechanism analysis of microglia across species provides insights into the conserved TFs regulatory module for microglia evolution. Together, our study offers an important resource for a better understanding of pig biology, xenotransplantation, evolution, development, and regenerative medicine research.

## Methods

**Ethical statement**. The study was approved by the Institutional Review Board on the Ethics Committee of BGI (Approval letter reference number BGI-NO.BGI-IRB18135-T1). All experimental procedures were conducted following the national and institutional guidelines of using the experimental animals for research. All the applicable institutional and national guidelines for the care and welfare of animals have been strictly followed for the tissue sampling procedures.

**Tissue dissociation and sample preparation for scRNA-seq library generation**. Porcine tissues for scRNA-seq library generation were collected from a local slaughterhouse (*Sus scrofa domesticus*, three-way hybrid of Landrace, Large White and Duroc, age 6 months, Hårby Slagteren IvS). The fresh tissues were collected, immediately placed on ice, and processed within 30 min. Each tissue was dissociated and digested independently. To ensure efficient digestion and cell viability, after tissue dissociation (as described for each organ below), cell suspensions were filtered by cell strainers after debris removal and red blood cell lysis. The cell viability (Hoechst 33342 (Invitrogen, Cat#H3570) and Propidium Iodide (Thermo Fisher Scientific, Cat#P3566)) from each tissue were evaluated by flow cytometry (FC) analysis on a NovoCyte Quanteon analyzer (Acea bioscience, Inc, US) provided by the FACS Core Facility, AU.

*Liver*. The fresh liver was collected and sampled with five different anatomical regions: left lateral lobe (LLL), left medial lobe (LML), right medial lobe (RML), right lateral lobe (RLL), and quadrate lobe (QL). For each region, 1 g tubes were punched and washed twice with cold PBS. The samples were mixed and carefully dissected into small pieces, and then transferred into a 50 mL tube with 20 mL digestion medium containing 0.5 mg/mL collagenase type II (Gibco,

Cat#17101015), 1.25 mg/mL protease (Sigma, Cat#P5147-100MG), 7.5 µg/mL DNase I (Sigma-Aldrich, Cat#D4527-10KU) in cold HBSS. The tissue digestion was performed at 37 °C for 15 min with gently shaking once every 5 min. The reaction was stopped in 20 mL cold MACS buffer containing 0.25% BSA (Sigma-Aldrich, Cat#10735096001), 2 mM EDTA, in PBS, and then filtered with 100 µm cell strainer (Sigma-Aldrich, Cat#CLS431752-50EA), and the cells were stained for the FC analysis.

*Spleen*. The fresh spleen was collected and sampled with 1 g pieces for both anatomical sides including the parietal and visceral side and washed twice in cold PBS. The samples were mixed and carefully dissected into small pieces and transferred into a 50 mL tube with 10 mL digestion buffer containing 20 mg/mL collagenase IV (Gibco, Cat#17104019), 1 U/mL Dispase II (Gibco, Cat#17105041), 7.5 µg/mL DNase I in 10 mL HBSS. The tissue digestion was performed at 37 °C for 15 min with gently shaking once every 5 min. The reaction was stopped in 20 mL cold MACS buffer containing 0.25% BSA, 2 mM EDTA, in PBS, and filtered with 100 µm and 40 µm cell strainer, and the cells were stained for the FC analysis.

*Retina*. The porcine eyes were collected and dissected by surgical scissors, and the retina was transferred to cold HBSS by using forceps. Subsequently, the retina was dissected into small pieces and put in 10 mL digestion buffer with 2 mg/mL collagenase I (Gibco, Cat#17018029), 3.75 µg/mL DNase I into 10 mL HBSS. The tissue was digested at 37 °C for 20 min with gently shaking once every 2 min. The digestion reaction was stopped in 20 mL MACS buffer with 0.25% BSA, 2 mM EDTA in PBS and filtered with 40 µm cell strainer, and the cells were stained for the FC analysis.

*Brain*. The fresh porcine brain was collected including 7 different regions: neocortex, cerebellar cortex, caudate nucleus, thalamus, hippocampus, hypothalamus, and pons with 0.5 g pieces for each region. The samples from different regions were mixed and dissected into small pieces. The digestion was performed at 37 °C for 30 mins in 20 mL digestion buffer containing 10 mg/mL collagenase IV, 15 µg/mL DNase I, followed by gently shaking once every 5 min. Subsequently, the digestion reaction was stopped by using MACS buffer and filtered with a 40 µm cell strainer. After centrifuge and resuspension, the cells were stained for the FC analysis.

*Lung*. Seven different regions: left apical lobe, left middle lobe, left main lobe, right apical lobe, right middle lobe, accessory lobe, and right main lobe were collected from the fresh porcine lung. From each region, a 0.5 g piece was collected and washed twice in cold HBSS. The samples were dissected into small pieces and put into 50 mL Falcon tube with 20 mL digestion medium containing 1 mg/mL collagenase type II, 2.5 mg/mL collagenase type IV, 7.5 µg/mL DNase I. The samples were digested at 37 °C for 30 min with gently shaking once every 5 min and stopped digestion in MACS buffer. The samples were diluted into cold HBSS and then filtered with 100 µm and 40 µm cell strainers respectively. Subsequently, the debris was removed, and the cells were stained for the FC analysis.

*Visceral adipose tissue and subcutaneous adipose tissue*. For each of the adipose tissue depot, 10 g of each tissue was weighed and placed in the 25 mL digestion media containing: FBS-free KnockOut™ DMEM medium (Gibco, Cat#10829018) supplemented with 1% (v/v) Penicillin/Streptomycin (Thermo Fisher Scientific, Cat#15140122), 2 × Antibiotic-Antimycotics (Thermo Fisher Scientific, Gibco #1524-062), 1 mM Sodium Pyruvate (Thermo Fisher Scientific, Cat#11360070), MEM Non-Essential Amino Acids Solution (MEM-NEAA) (Thermo Fisher Scientific, Cat#11140035), 2 mM L-Glutamine (Thermo Fisher Scientific, Gibco #25030-024), 0.2% collagenase type I, 0.25 U/mL Dispase II and 7.5 µg/mL DNase I. Samples were incubated at 37 °C in a water bath for at least 45 min, shaken and mixed by pipetting every 5 min. At the end of the incubation time, a PBS-based wash buffer containing 0.5% BSA, 2 mM EDTA in PBS was added and the cell suspension was filtered through a 100 µm cell strainer, and centrifuged at 300 g for 7 min. The washing step was repeated 2 times more and the cell suspension was always re-filtered through 100 µm cell strainer and transferred to a new canonical tube to remove the access of undigested clumps and fat.

*Intestine*. The fresh intestine was collected and washed twice in cold PBS. 3 g of intestine pieces were picked and dissected into small pieces. The digestion reaction was at 30 min with gently shaking once every 5 min in 20 mL digestion medium containing 5 mg/mL collagenase type I, 2.5 mg/mL collagenase type IV, 15 µg/mL DNase I. After digestion, the tissues were diluted into cold HBSS and then filtered with 100 µm and 40 µm cell strainers respectively. The debris was removed, and the cells were stained for FC analysis.

*PBMC*. The approaches for porcine PBMC isolation were based on the human PBMC isolation protocol in our previous study[102]. In brief, 10 mL of pig blood sample in a citrated vial was gently inverting to mix well. The PBMC were isolated followed by density gradient centrifugation with Ficoll®-Paque Premium medium (Cytiva, Cat#17-5442-02). The cells were resuspended after red blood cells lysis and stained for the FC analysis.

**Cultured ECs isolation from pig lung**. One leaf of pig lung was used for perfusion with PBS from the big vessel on top. After perfusion, the lung was squeezed and extracted the PBS. Fifteem milliliters of the digestion buffer containing 0.1% collagenase II, 0.25% collagenase IV, 75 µL DNASe I in KnockOut™ DMEM medium was injected into the sealed pig lung. The opening site was closed by sealing clip and left in the bag into a water bath at 37 °C for 30 min. Next, the lung was taken out and the digestion buffer has flowed away. Twenty milliliters of PBS was used to wash the lung thrice, and the medium was collected for filtering with 100 µm cell strainer afterward. The medium was transferred into 10 mL Falcon tube and centrifuged at 300 × g for 5 min. The cells were washed with 10 mL PBS and centrifuged for culturing into 5 mL PAEC medium (DMEM with 10% fetal calf serum, 1 × MEM-NEAA, 1 mM sodium pyruvate, 1 × glutamax, 1 × penicillin/streptomycin) in $CO_2$ incubator.

**Cultured ECs isolation from pig aorta**. The aorta near to lung was separated and washed with PBS. Five milliliters of digestion buffer was injected into the sealed aorta and left the aorta into a water bath at 37 °C for 30 min. Next, the aorta was taken out, and the digestion buffer was flowed out. The medium was transferred into a 15 mL Falcon tube and centrifuged at 300 × g for 5 min. The collected cells were washed with 5 mL PBS and centrifuged for culturing within 5 mL PAEC medium.

**Single-cell library construction and sequencing**. Library generation: Single-cell RNA-sequencing (scRNA-seq) libraries were prepared following the manufacturer's instructions of GemCode Single Cell 3′ Gel Bead and Library kit (v3 Chemistry) from 10× Genomics, Inc. (Pleasanton, CA). Briefly, scRNA-seq library was generated by the cells from the different tissues: lung, liver, intestine, spleen, adipose, brain, and retina. After library construction, the library conversion was performed using the MGIEasy Universal DNA Library Preparation reagent kit (BGI, Shenzhen, China) for compatibility, followed by sequencing on a DNBSEQ-T7 platform (MGI).

**Sample collection for nucleus extraction and snRNA-seq**. The approaches for sample collection and nuclei extraction for snRNA-seq libraries generation in this study were followed in our previous study[28]. Briefly, the tissues used in this study: heart, kidney, spleen, liver, lung, retina, and brain regions (area_postrema, cerebellum, subfornical organ, and OVoLT) were carefully dissected from the healthy domestic pigs (*Sus scrofa domesticu*s, three-way hybrid of Landrace, Large White and Duroc, age 3 months) with strict compliance to the ethical guidelines. The collected tissues were washed with cold PBS, and immediately frozen in liquid nitrogen and stored in a −80 °C freezer before use. For the nuclei extraction process, the tissues were thawed and cut into small pieces, then transferred into a homogenization buffer containing 20 mM Tris pH 8.0, 500 mM sucrose, 0.1% NP-40, 0.2 U/mL RNase inhibitor, 1% BSA, and 0.1 mM DTT. The tissue pieces were grinded for 15 times with tight pestles and filtered with 40 µm strainer. The samples were centrifuged at 500 × g for 10 min at 4 °C to carefully discard the supernatant. The pellets (nuclei) were resuspended in PBS containing 1% BSA and 20 U/µL RNase Inhibitor for later snRNA-seq library construction (MGI).

**Single-nuclei library construction and sequencing**. The mRNA capture was operated on a DNBelab C4 device (MGI). cDNA amplification and libraries construction were generated using the MGI DNBelab C series reagent kit (MGI) following the manufacturer's instructions. All the libraries were sequenced on the DNBSEQ-T7 platform.

**Pre-processing and quality control of scRNA-seq and snRNA-seq data**. Cell Ranger 3.0.2 (10x Genomics) was used to process the raw sequencing data of scRNA-seq. The sequencing data from snRNA-seq was filtered, and the gene expression matrix was obtained using DNBelab C Series scRNA analysis software (MGI). The reference genome was downloaded from the Ensemble assembly: Sscrofa11.1. Cells were only retained if the number of detected genes were greater than 200 and less than 5000 and the percentage of detected mitochondrial transcripts from MT genes (ATP6, ATP8, COX1, COX2, COX3, CYTB, ND2, ND3, ND4, ND4L, ND5, ND6) was less than 30%. The Pig ND1 gene was not included in MT-based filtering due to high sequence variant in pigs.

**Identification of cell clusters**. After filtering, unsupervised clustering was performed using Seurat v3[32]. Datasets from different sequencing libraries underwent normalizing and scaling. Variable genes were determined using Seurat's "FindVariableGenes" function with default parameters (selection.method = "vst", nfeatures = 2000). Clusters were identified via the "FindClusters" function (0.8 < resolution < 1.5) implemented in Seurat using principal components with a P value < 0.01 and subsequently visualized using the "RunTSNE" and "RunUMAP" functions (reduction = "pca").

**Identification of differentially expressed genes (DEGs) across clusters**. "FindAllMarkers" function implemented in Seurat v3 was used to identify DEGs across clusters with the options "min.pct = 0.25, logfc.threshold = 0.25". Multiple test correction for *P* value was performed using the Bonferroni method, and 0.05 was set as a threshold to define significance. Furthermore, cell-type identities were assigned using canonical cell-type markers.

**Gene ontology (GO) enrichment analysis**. Gene Ontology (GO) analysis was using in the clusterProfiler 4.0 package[103]. The GO terms of selected genes were enriched in the database "org.Hs.eg.db" using "enrichGO" function because of the lack of study in pigs. Benjamini–Hochberg (BH) method was used for the multiple test adjustment.

**Integration of multiple datasets from different sequencing platforms**. Datasets derived from the 10X (scRNA-seq) and the DNBelab C4 (snRNA-seq) platforms were integrated using the Seurat R package (version 3.2.2) after cell filtering. In detailed, data was log1p-normalized with the SCT normalizeData method using the "SCTransform" function, and subsequently scaled by the Pearson Residuals with a scale factor of 10,000 as default using the "ScaleData" function. The top 3000 highly variable features were selected using the "SelectIntegrationFeatures" function, followed by finding the integration anchors using the "FindIntegrationAnchors" function, performing the integration of the data using the "IntegrateData" function. Following integration, principal component analysis was performed using the "RunPCA" function with default parameters, then both t-SNE (t-Distributed Stochastic Neighbor Embedding) and UMAP (Uniform Manifold Approximation and Projection) dimensionality reduction methods were conducted based on the top 20 principal components (PCs) using the "RunTSNE" and "RunUMAP" functions, respectively. Moreover, unsupervised clusters were identified by setting the top 20 PCs and a clustering resolution of 1.0 using "FindNeighbors" and "FindClusters" functions. The Pearson correlation coefficients of cell types were calculated using the average expression of top 3000 highly variable features and visualized using pheatmap R package (1.0.12).

**Pseudotime trajectory analysis**. *PECAM1 + PTPRC-EPCAM-COL1A1-PDGFRB-HBB*^low ECs was used to subset the ECs from adipose tissues. Then, the "IntegrateData" function with cca methods in Seurat package were used to correct the batch effect of ECs in adipose-V and adipose-S. The integrated ECs of both adipose-V and adipose-S were clustered by "FindClusters" with resolution 1.0. Each cluster was annotated based on canonical subtype markers and cluster specific DEGs. Both all ECs and EndMT subtype were used to perform the subsequent pseudotime trajectory analysis. Monocle 2[104] package was used to discover the cell state transitions ECs. Genes expressed in less than 10 cells were filtered out. DEGs were computed by function "differentialGeneTest" in monocle2. Genes with qvalue less than 0.01 were regarded as DEGs and sorted by qvalue using "setOrderingFilter" function. The pseudotime trajectory was constructed by "DDRTree" algorithm with default parameters. The dynamical expression changes of selected marker genes by pseudotime were visualized by "plot_genes_in_pseudotime" and "plot_pseudotime_heatmap" function.

**Cross-species comparison of intercellular communications between pig and human**. Intercellular communication analysis was conducted using CellChat (v0.0.1) R package with default parameters[65]. Pig and human liver, kidney, and heart datasets were analyzed separately. The human liver[105], kidney[106], and heart[107] datasets were collected from previous reports. Intercellular communications analysis was performed based on cell types shared by the separate liver, kidney, and heart between pig and human. Cell–cell communication network was visualized using the "netVisual_aggregate" function, centrality score was computed and visualized using the "netAnalysis_signalingRole_network" function, relative contribution of each ligand-receptor pair was visualized using the "netAnalysis_contribution" function.

**TF-target interaction inference**. TFs gene list was downloaded from the animalTFDB3.0[108]. Only genes expressed in more than 5% of corresponding cell types were subjected to GENIE3[109] (v1.8.0) to infer putative regulatory circuits from expression data using tree-based ensemble methods. TF-target pairs with weight value more than 0.01 were retained for primary regulatory network construction. The total frequency of each TF-target pair present in 13 species was counted to evaluate its conservation level. TF-target pairs present in at least 5 species were considered as 'conserved regulomes'. Conserved regulomes were visualized using the igraph[110] (v1.2.6) R package.

**Construction of the PCA database**. The PCA database was generated with ShinyCell[111] with default settings and modified to include the introduction and user guide pages.

**Tissue processing**. All peripheral pig tissues were stored in 70% ethanol at 4 °C. The pig brain tissues were stored in PBS buffer at 4 °C and changed into 70% ethanol one week prior to paraffin embedding. We first dehydrated the tissues with

absolute alcohol (VWR chemicals) and xylene (Histolab). Next, paraffin (Histolab) immersion was performed using an automated Tissue Processing Center TPC 15 Duo (MEDITE) machine. After manual embedding into separate paraffin tissue blocks, one representative section (4um) was taken from each tissue block using a microtome (Microm HM 355 S, Thermo Fisher Scientific). A microm STS Section-Transfer-System (waterfall) was used for section transfer to a warm water bath (38 °C) stretching before placed on SuperFrost PlusTM slides (VWR). All slides were dried at room temperature for 24 h followed by 50 °C overnight (LAMB Windsor Incubator E18.31, Histolab).

**Immunohistochemical staining**. Antibodies produced within the HPA project (see Supplementary Data 14 for antibody information) with high reliability (based on human antibody validation[112,113]), and over 80% sequence homology to pig orthologs was used for immunostaining. All antibodies are published on the HPA portal (www.proteinatlas.org) with more details about antibody reliability and tissue distribution in humans Formalin-fixed paraffin-embedded (FFPE) pig tissues, previously validated and prepared[30], was utilized for validating that scRNA-seq is a robust method for the classification of different cell types. The staining protocol follows our previous study by Karlsson et al, as briefly described below. Full size tissue sections as well as sections from tissue micro array (TMA) were represented by 1 mm punches moved from the donor block to an empty recipient paraffin block. Deparaffinization and rehydration were performed by Autostainer XL (ST5010, Leica biosystems) followed by heat-induced epitope retrieval in pH6.1 citrate buffer (DAKO, diluted 1:10 with deionized water) and pressure boiler (decloaking chamber, Biocare Medical). Autostainer 480 (ThermoFisher Scientific) was used for automated IHC staining with UltraVision™ Quanto Detection System HRP DAB-kit from Thermo Fisher Scientific including Ultra V Block, HRP Polymer, Primary Antibody Enhancer, DAB Quanto Substrate, DAB Quanto Chromogen, and primary antibodies were diluted in Antibody Diluent OP Quanto. After a 5 min block (Ultra V Block), primary antibodies were incubated for 30 min, followed by the secondary HRP Polymer (ThermoFisher Scientific) and the final step of 5 min DAB Quanto incubation. All steps were separated by double wash (Tris Buffer and Tween 20). Slides were placed in water and moved to the Autostainer XL (ST5010, Leica biosystems) for counterstaining (hematoxyline), dehydration, and cover glass mounting. Image digitalization was performed with Scanscope AT2 (Aperio) using a 20× objective.

**Immunofluorescence staining**. Two slides with adipose tissue samples were dewaxed using Histoclear. One slide of adipose tissue was then incubated with ACTA2 monoclonal mouse antibody (1:100 dilution) and PECAM1 polyclonal rabbit antibody (1:50 dilution) overnight at 4 °C. The other slide was incubated with VWF monoclonal mouse antibody (1:7,000 dilution) and TAGLN polyclonal rabbit antibody (1:100 dilution). The slides were then washed in TBS-Tween and blocked with TNB buffer. To visualize the proteins, an HRP-conjugated swine anti-rabbit antibody (diluted 1:200 in TNB buffer) was added and the slides were incubated at room temperature for 30 min. The slides were washed and then incubated with TSA-FITC substrate for 15 min in the dark. After this step, the slides were placed in 0.1% NaN3 in PBS to inactivate the secondary antibody. They were then incubated in the dark with an HRP-conjugated donkey anti-mouse antibody (diluted 1:200 in TNB buffer), and after washing incubated with a TSA-Cy3.5 substrate.

**TGF-β2 treatment-induced EndMT**. Primary pig aortic endothelial cells (PAECs) were cultured in Dulbecco's modified Eagles's medium (DMEM) with 4.5 g/L D-glucose, L-Glutamine, and pyruvate (Gibco, #41966052), supplemented with 10% fetal bovine serum (FBS) (Sigma, #F7524), 1× non-essentials AA (MERCK, #M7145), and 1× penicillin/streptomycin. The PAECs were cultured on 0.1% gelatin (MERCK, #D8537) coated culturing flasks (Nunc, #156499) in a 5% $CO_2$ humidified incubator at 37 °C.

Primary human umbilical vein endothelial cells (HUVECs) were cultured on 0.1% gelatin-coated culturing flasks in M199 medium (Gibco, #22350-029) supplemented with 2mM L-Glutamine (Gibco, #35050-061), Endothelial Cell Growth Supplement (ECGS)/ Heparin (PromoCell, #C-30120), and 20% FBS, in a 5% $CO_2$ humidified incubator at 37 °C. The M199 medium with supplements was replaced three times per week. HUVECs were passaged every 14 days by washing the cells twice with PBS, trypsinized, and centrifuged at $250 \times g$ for 5 min.

For TGF-β2 treatment, 100,000 PAECs/well were seeded in 12-well plates (Thermo Scientific, #150628) supplement with 2 ng/mL TGF-β2(Merck, T2815) in triplicates. Control cells were cultured in normal PAECs medium without TGF-β2. On day 2, the PAECs were trypsinized and seeded (100,000 PAECs/well) in 12-well plates followed by changing the medium on day 4. For HUVECs, which grow much slower compared to PAECs, cells were cultured in control (0 ng/mL, TGF-β2) and TGF-β2 medium (2 ng/mL) for 5 days without passaging. All cells were harvested for analysis of ACTA2 and CD31 expression by flow cytometry 5 days after TGF-β2 treatment.

**Flow cytometry analysis of ACTA2 and CD31 expression**. Cells (PAECs and HUVECs) were washed in PBS twice, dissociated with trypsin (Gibco, #25300054), and centrifuged at 400 g for 4 min. The cell pellets were washed with PBS + 5% FBS

twice and resuspended in 240 μL PBS + 5% FBS. PAECs were stained with ACTA2 (AF488 Human Alpha-Smooth muscle actin, R&D, Cat#IC1420G, 1:100 dilution), CD31 (Porcine CD31/PECAM1 AF700, R&D, #FAB33871N-100UG, 1:100 dilution), and CD45 (BV421 Mouse Anti-Human CD45, BD Bioscience, #563879, 1:100 dilution). HUVECs were stained with CD31 (FITC Mouse Anti-Human CD31 (BD Bioscience, #555445), 1:100 dilution) and ACTA2 (AF488 Human alpha-Smooth muscle actin, R&D, Cat#IC1420G, 1:100 dilution) separately. Cells were incubated on ice in the dark for 30 min. Before FC analysis, the stained cells were washed twice with pre-cooled PBS + 5% FBS. At the final wash, the cells were resuspended in 200 μL PBS + 5% FBS. Prior to this study, we compared the pig and human ACTA2 amino acid sequences, which are identical. Antibody validation of anti-ACTA2 was also validated by staining human mesenchymal stem cells. Flow cytometry was performed with the NovoCyte Quanteon analyzer (Acea bioscience, Inc, US) provided by the FACS Core Facility, Aarhus University.

**Statistics and reproducibility**. Statistical significance of differential expression gene was performed with multiple test correction for $P$ value (Bonferroni). Benjamini–Hochberg (BH) method was used for the multiple test adjustment for the gene ontology enrichment analysis. Unpaired, two-sided $t$-test was used for comparison of CD31 and ACTA2 expression in the induced EndMT experiments in cultured ECs. A $P$ value less than 0.05 was considered statistically significant. Unless stated elsewhere, all immunohistochemistry, immunofluorescence staining and FACS experiments were performed with at least three experimental replicates. No statistical method was used to predetermine sample size. No data was excluded from the analyses. Filtering criteria for the low-quality cells and potential doublets are provided in the method above and the source code. Randomization is not related to this study. The investigators were not blinded to allocation during the experiments and the outcome assessment.

**Reporting summary**. Further information on research design is available in the Nature Research Reporting Summary linked to this article.

## Data availability
The single-cell and single-nuclei RNA-sequencing data generated in this study have been deposited in the CNGB Sequence Archive (CNSA) of China National GeneBank DataBase (CNGBdb) under accession code "CNP0002165". The single-cell and single-nuclei RNA-sequencing data generated in this study have also been deposited in the gene expression omnibus database (GEO) under accession codes "GSE196055" and "GSE193975 [https://www.ncbi.nlm.nih.gov/geo/query/acc.cgi]". All matrix data can be downloaded from the PCA database (https://dreamapp.biomed.au.dk/pigatlas/). All other relevant data supporting the key findings of this study are available within the article and its Supplementary Information files or from the corresponding author upon reasonable request. Source data are provided with this paper.

## Code availability
All codes for processing of the single-cell RNA-sequencing data have been deposited to GitHub[114] and available through this URL: https://github.com/Dingpw/PigAtlas.

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

## Acknowledgements

We thank the FACS core team from the Department of Biomedicine at Aarhus University for the help of FACS experiments and data analysis. We thank Trine Skov Petersen and Xi Xiang for their technical assistance. This project was partially supported by the Sanming Project of Medicine in Shenzhen (SZSM201612074, to L.B. and Y.L), Guangdong Provincial Key Laboratory of Genome Read and Write (No. 2017B030301011 to X.Xu.), Guangdong Provincial Academician Workstation of BGI Synthetic Genomics (No. 2017B090904014 to X.Xu.), CAMS Innovation Fund for Medical Sciences (CIFMS) (22021-I2M-1-061 to D.C.), the DFF Sapere Aude Starting grant (8048-00072A to L.L.), and the Novo Nordisk Foundation (NNF21OC0071718 to L.L.). J.K. is supported by AIAS-CO-FUND II: GA: MSCA:754513, Lundbeckfonden: R307-2018-3667, Carlsberg Fonden: CF19-0687, Kræftens Bekæmpelse: R302-A17296, A.P. Møller Fonden: 20-L-0317, Riisfort Fonden and Steno Diabetes Center Aarhus (SDCA). P.C. is supported by Grants from Methusalem funding (Flemish government), the Fund for Scientific Research-Flanders (FWO-Vlaanderen), the European Research Council ERC Advanced Research Grant EU- ERC74307 and the NNF Laureate Research Grant from Novo Nordisk Foundation (Denmark). Y.L. is supported by the Independent Research Fund Denmark (9041-00317B), European Union's Horizon 2020 research and innovation program under grant agreement No 899417, and the Novo Nordisk Foundation (NNF21OC0068988; NNF21OC0071031). We thank Hårby Slagteren IvS for providing the study materials. We thank the China National GeneBank for the support of executing the project under the framework of Genome Read and Write. We thank Professor Fred Dubee for providing critical comments and language editing for the manuscript.

## Author contributions

Conceptualization, Y.L., D.C., L.L., M.U., L.B., and H.Y.; Methodology, F.W., P.D., X.L., X.D., C.B.B, and E.S.; Investigation, F.W. (major), P.D. (major), X.L., X.D., C.B.B, E.S., S.B., J.Z., L.Z., L.P.D.R., Li.L, Y.W., W.Z., Z.L., J.H., R.L., Q.Q., Y.J., W.W., Y.Y., M.P., H.W., A.W., Jac.H., J.K., and M.K.; Sequencing, L.X., P.L., F.C., and H.J.; Writing – Original Draft, F.W. and Y.L.; Writing – review & editing, all authors; Funding acquisition, M.U., L.B., H.Y., X.X., D.C., L.L., and Y.L.; Resources, X.Z., R.B.H., L.F., C.L., F.P., M.U., L.B., N.J., X.X., H.Y., P.C., J.M., D.C., L.L., and Y.L.; Supervision, X.Z., R.B.H., L.F., C.L., F.P., M.U., L.B., N.J., X.X., H.Y., P.C., J.M., D.C., L.L., and Y.L.

## Competing interests

The authors declare no competing interests.

## Additional information

[1]Lars Bolund Institute of Regenerative Medicine, Qingdao-Europe Advanced Institute for Life Sciences, BGI-Qingdao, BGI-Shenzhen, Qingdao, China. [2]Department of Biomedicine, Aarhus University, Aarhus, Denmark. [3]BGI-Shenzhen, Shenzhen, China. [4]College of Life Sciences, University of Chinese Academy of Sciences, Beijing, China. [5]Department of Biology, University of Copenhagen, Copenhagen, Denmark. [6]Steno Diabetes Center Aarhus, Aarhus University Hospital, Aarhus, Denmark. [7]Department of Neuroscience, Karolinska Institutet, Stockholm, Sweden. [8]MGI, BGI-Shenzhen, Shenzhen, China. [9]Laboratory of Angiogenesis and Vascular Metabolism, Center for Cancer Biology, VIB, Leuven, Belgium. [10]Department of Oncology, Leuven Cancer Institute, KU Leuven, Leuven, Belgium. [11]College of Basic Medicine, Qingdao University, Qingdao, China. [12]School of Basic Medical Sciences, Binzhou Medical University, Yantai, China. [13]Institute of Systems Medicine, Chinese Academy of Medical Sciences, Peking Union Medical College, Beijing, China. [14]Suzhou Institute of Systems Medicine, Suzhou, China. [15]Aarhus University of Advanced Studies (AIAS), Aarhus University, Aarhus, Denmark. [16]Department of Protein Science, Science for Life Laboratory, KTH-Royal Institute of Technology, Stockholm, Sweden. [17]Department of Obstetrics and Gynecology, Aarhus University Hospital, Aarhus, Denmark. [18]Department of Immunology, Genetics and Pathology, Uppsala University, Uppsala, Sweden. [19]IBMC-BGI Center, the Cancer Hospital of the University of Chinese Academy of Sciences (Zhejiang Cancer Hospital), Institute of Basic Medicine and Cancer (IBMC), Chinese Academy of Sciences, Hangzhou, China. [20]Center for Biotechnology, Khalifa University of Science and Technology, Abu Dhabi, United Arab Emirates. [21]These authors contributed equally: Fei Wang, Peiwen Ding, Xue Liang, Xiangning Ding, Camilla Blunk Brandt. ✉email: cds@ism.pumc.edu.cn; lin.lin@biomed.au.dk; alun@biomed.au.dk

