## [Peer Review File · Nature Communications]

REVIEWER COMMENTS

Reviewer #1 (Remarks to the Author):

In this submission, Wang and colleagues present single cell/nucleus RNA sequencing analysis of 20 pig organs, with the aim of generating an atlas of gene expression across tissues and cell types. They then focus on dissecting endothelial cell heterogeneity and transcription factor regulation of microglia identity. Although the objective of the study is important because of the relevance of the porcine model to studies of human disease and in xenotransplantation, there are several concerns that limit the utility of this data with respect to defining endothelial heterogeneity and superficiality of the analyses.

Major comments:

1. Significant under-representation of endothelial cells in this dataset hampers investigation of their anatomic heterogeneity. Two other whole body single cell atlases (Tabula Sapiens and Tabula Muris) yielded substantively greater proportions of endothelial cells. It does not appear that sufficient endothelial cells were sequenced to provide an informative dataset to establish their transcriptional diversity. This was a limitation in Tabula Muris necessitating the endothelial-focused study by Kalucka et al.
2. In this atlas, the vast majority of endothelial cell events derive from adipose tissue, with very few from other highly vascularized organs which would be more relevant. Why is there a preponderance of adipose endothelial cells?
3. I could not determine from the methods what pipeline was used for cell type annotation? PECAM1 alone is not adequate to identify the endothelial lineage; it is expressed by other cell types, like monocytes, and it is lowly expressed by some endothelium, like liver sinusoids. Figure 3F shows that most of the other definitive markers of endothelial cells are in fact negative on the endothelial cells from all other organs. Could the authors please provide more information about how cell types were assigned?
4. How can the authors be confident that the EndMT population is not simply contamination of fibroblasts or smooth muscle cells? There is a clear inverse correlation between PECAM1 expression and ACTA2/TAGLN, and population 9 in Figure 4C and population 4 in Figure 4G show low to negative PECAM1 and CDH5 expression compared with all other subsets. From this random snapshot in time, without a lineage tracer, it is difficult to say that cells co-expressing two markers (one low, one high) truly represent a transitional cell state. What evidence other than low PECAM1 expression can the authors provide to demonstrate that these cells are endothelial in origin?
5. For the EndMT cells, could the authors also show expression of known regulators of this process, such as SNAI1, SNAI2, TWIST? Could they also please present a dot plot of co-expression of PECAM1-ACTA2, and other endothelial-mesenchymal genes?
6. The authors state that the data have been deposited into the CNGBdb, but I could not find the data in this database.
7. Ethics statement is a bit awkward: "All experimental procedures were conducted following the guidelines of the experimental animals." Can the authors reword this to more specifically indicate which animal welfare guidelines were followed?
8. The authors conclude their manuscript with a statement that essentially invalidates their approach: "when the number and composition of cell types are essential, we recommend

using only one method...consistently to avoid method-induced biases." One could argue that the number and composition of cell type ARE essential in this study. Therefore, could the authors better justify their approach in light of this?

Minor comments:

1. Page 5, line 161: "sing-cell" should be "single-cell"
2. Page 9, line 319: Should be "co-expression" rather than "co-expressing"
3. Page 14, line 491: I believe the authors mean attachment rather than "detachment."
4. Page 14, line 518: "In addition, pathological angiogenesis in the development of human disease such as cancer." This is an incomplete sentence.
5. Page 20, line 718: "To further ensure the quality of the dataset." This is an incomplete sentence.
6. The authors note that HLA-DRA was expressed by a subset of endothelial cells (Figure 3C). Should not the gene name be SLA-DRA (Swine Leukocyte Antigen).
7. ICAM2 is rather a better marker of endothelial cells than ICAM1, which is expressed by many other cell types (Figure 4E).
8. Axes are very small and illegible in Figure 2.

Reviewer #2 (Remarks to the Author):

#SUMMARY

Wang et al generated a single cell transcriptome atlas of many pig tissues using both single cell and single nuclei methodology.

After performing adequate QC of the data, including comparison of single cell and nuclei data for a few of the tissues, they identified multiple known cell types and identify novel endothelial cell states.

After the initial broad atlas analysis, they focus on:

-Identifying multiple sub-states for endothelial cells, and validated the proposed endothelial-mesenchymal state in vitro

-Then, cell-cell interaction analysis is performed for endothelial cells

-Finally, they perform an evolutionary analysis of microglia, and apply gene regulatory network analysis to find that MEF2C in a conserved "region" of these cells.

-In order to provide our best feedback and help authors detect inconsistencies, we re-ran the analysis using their provided processed files; we appreciate that the authors provided this data in a timely manner.

Overall, this is a well done atlas with interesting findings. It will be an useful resource for the research community. However, like many other single cell atlas studies, the manuscript is affected by perhaps too broad a scope, and lack of focus.

#MAJOR COMMENTS

-By including the description of general findings (e.g., in X tissue, we found X cell types expressing Y and Z), the manuscript becomes long, hard to follow, and dilutes the main findings that are discussed later. I think this dilutes the impact of the later findings. This is a common issue with most single cell atlas publications, and I don't have a great solution other than perhaps making the first few sections shorter and with less detail.

-Then, the two main in depth analyses (the endothelial cell heterogeneity, and the microglia

evolution and GRN) are shown, which are great but seem disconnected. Perhaps by shortening the initial general description sections the manuscript will be more readable. Otherwise, provide rationale as to why those two were selected for this manuscript.

-Please deposit data (raw and processed files) in GEO and provide accession number, add statement in manuscript with this information.

-We could not find three of the samples mentioned in the metadata (lung samples with sample IDs "PG-LG-1", "PG-LG-2", and "PG-LG-3"). We found 219,140 cells after passing their QC filter whereas the paper mentions 222,526 cells (i.e. I have 98% of the cells they mention). Please correct this so that data and manuscript match.

-Provide reasoning to combine single cell and single nuclei approaches in this analysis, other than "practicality and resource availability"

-It is unclear how many animals were sampled, and which tissues/cells are from which animals, and how they may be from the same animal or not. The metadata should include this, and include a table with this data (can be in supplement). Batch effects from sampling different animals can result in false "cell states" in clustering analysis (e.g. if all cells from a "novel state" come from a single animal), this affects the the conclusion of existence of Endothelial-mesenchymal transition cells.

-p9, lines 316-318: It would be helpful to show co-expression in the single cell RNA-seq data in addition to IF images. When we filtered only the cells that were PECAM+/PTPRC-/EPCAM-, there was a cluster of cells that were ACTA2+/TAGLN+ and a cluster of cells that were CD68+/C1QB+. Is it possible that these cells are actually smooth muscle cells? Similarly, is it possible that the "immune-activated" endothelial cells are macrophages? Please provide evidence using the scSeq data that these cells are distinct from smooth muscle and macrophages, respectively. For example, comparing the clusters against existing cell type databases (rather than using manually picked markers) may add further evidence of correct identification of cell clusters/types.

-p10, lines 333-339: It does not seem we have access to the cultured primary ECs in the processed data. Please make sure raw and processed files are available in the deposited data.

-p11, line 375: cell communication analysis does not prove that the cell-cell interaction is actually happening, it only suggests hypotheses to directly test; please reflect this in the writing, for example: "The analysis suggests that the VEGF signaling pathway is used by most renal cells for intercellular communication"

#MINOR COMMENTS

-p3 line 75: Merge two sentences for conciseness: However, several species-specific cellular and molecular differences between pigs and humans exist; for example, pluripotent progression and metabolic transition were found to be different using single cell RNA-sequencing (scRNA-seq) (Liu et al., 2021)

-p3 line 87: Make sentence clearer, no need to list all species: "Pioneering work has been completed for most model animals and humans (References)"

-p4 line 114: change heading to "single-cell and single-nuclei RNA sequencing of four pig tissues highly correlates in common cell types"

-p5 line 152: "also identified from the two different methods"

-p7 line 254: remove "much" -> have been extensively studied...

-p12 line 419: I think you mean "inference" rather than "interference"

-Figure 1 inset (4) of Bioinformatic analysis says "virus receptor" and has a little diagram, but this was not included in this study, recommend removing

-Figure 2 panel B: It appears that the heat map data (here and in other heat maps too) has been blurred during figure generation. One possible cause is the file format used to save, which sometimes causes this in heat maps. Please correct.

Reviewer #3 (Remarks to the Author):

Review NCOMMS-21-46827-T

General comments

This manuscript reports single cell RNA seq data from approximately 20 adult tissues from the pig, with about 2/3 of the data coming from single cell and 1/3 coming from single nuclei technologies. This is an important resource for the scientific community interested in this species as a model for human, as well as investigations into the pig itself. The authors document similarities and differences between tissues at the single cell level and relationships at the transcriptome level across each tissue. They use as an example retina and kidney and document and verify these expression patterns using protein-based technologies. They then focus on endothelial cells and they provide some evidence for an intermediate stage cell type that is transitioning from an endothelial cell type to a mesenchymal cell type. They compare interacting pathways between human and pig for liver, heart and kidney and then finish the manuscript with a comparison across many species for the differences in gene expression in regulatory pathways present in microglia. The data analysis appeared sound, but additional details on the methodology are needed. Most importantly, they need to include more information on integration of the data, including normalization and scaling. This is essential for any single-cell data but especially important for projects integrating scRNA-seq and snRNA-seq. For the identification of cell clusters, they should explicitly state how many principal components were included and the resolution used for clustering. While there are a number of places in the manuscript that should be revised, overall the paper it does a very good job of describing a complex and deep data set that will be of value to many working in this species.

The major concerns are described in detail below, but include the low evidence for a clear transitioning population of EC to M cells, and the apparently simplistic description of the CellChat results. It would strengthen both of these results if at least some component was further documented or validated through statistical analysis or protein-level experimentation.

Specific comments

I. 131. The filter for removing cells only if they have $\geq 30\%$ mitochondrial transcripts is very high, and is concerning. 30% is a very high cutoff for mitochondrial reads for all tissues. There is precedence in using 30% for high energy tissues such as heart. However, the authors should explain why this metric was chosen, and if it is appropriate to use for all tissues. The authors should also state what proportion of cells are in the dataset with $>10-20\%$ mito, which is more often the cutoff used.

I. 141-142. The data in S2 should be formatted to show a direct comparison between the two sequencing methods.

I. 145-146. It is difficult to see the different groups of cells sequenced in the tSNE plots. This is important to see how the data is impacted by sequencing method. For clarity, it might be better to show each group in its own tSNE plot.

I. 165-166. Figures 1D & 1E show the same data.

I. 269-276. They suggest identification of "immunomodulating ECs" in their EC population given expression of several macrophage markers. Because in line 260 they indicate immune cells can express PECAM1, and the filter here was only to remove cells that express PTPRC, it is not at all clear to me that they actually excluded all macrophages from the EC pool (is it

possible there are PTPRC- low or -negative macrophages?). If not, then this is not evidence of such an immunomodulating EC. Additionally, the authors should explain what expression metric was used and if individual cells were screened or if they applied the metrics to clustered cells. If it was not individual cells, there could be non-ECs included in the EC groups, which could impact downstream findings of transitioning cells.

I. 338-348. I do not find these data to be highly convincing that they have clustered a transitioning cell type. They report in Figure 4C,E and describe that the pseudotime analysis shows a EMT using a set of genes that the trend line is almost completely flat for most genes. Only PECAM1/CDH5 shown any major expression changes across the x axis. The only evidence they have that there is a EMT cluster is that there are a few- very few- cells in black in 4E that are positive for PECAM1 (and even fewer for CDH5) and M markers such as ACTA2, etc. This is reflected in the very low numbers of cells reported in 4C that express the E markers in cluster 9, especially for CDH5. Similar complaints can be lodged for the work in aorta (Figure 4G), where PECAM1 and CDH5 expression is virtually non-existent in cluster 4. Is there any statistical test that can be run to document a transition is occurring, or to exclude the possibility that these cells are contaminants?

I. 354-363. please add a short description of the principle behind CellChat, as it is not clear what is being measured here. In line 353-355, you describe that you are looking at "communication between ECs and other cell types", and then compared RNAseq data between pig and human to "gain better understanding of the cell-cell interactions in these three organs between pig and human". What is a "shared cell type" (I. 359), as distinct from simply shared gene expression patterns? Perhaps the explanation of what constitutes a "shared cell type" would suffice, but there could be further clarification on why there was not more shared cell types identified between pig and human (5-6 shared between tissues seems low).

I. 365. Cellchat apparently equates finding evidence of expression of a gene expressing a known ligand in cell A with expression of a receptor for that ligand in cell type B as "communication" between these cell types. The authors here appear to believe this inference, and "investigated signaling interactions" across cell types and tissues, and compared the results for the two species. There are many caveats to this analysis, none of which the authors admit or attempt to validate in any way with other techniques (such as demonstrating the presence of the ligand at the receiving tissue, or even expression of the protein of either partner in the tissues). For example, many ligands are not secreted in concert with gene expression levels, due to posttranscriptional or post-translational regulation. Further, the lack of such inferences based on a lack of such co-occurrences does NOT demonstrate a difference between species. There may be other explanations, such as biases using human genome annotations for the pig. The authors should describe the fairly large assumptions in these analyses, as well as at least some actual validations of activity of the inferred pathways. There was quite a bit of divergence in the results between cell signaling in pigs and human. Would the authors expect to see more conserved patterns between the two species?

I. 419. As above, please describe in more detail the method that you are using for GRN analysis, as there are several methods, rather than simply say you ran an analysis using GENIE3. (and "interfering" presumably you mean "inferring"?). In this paragraph, the authors also use the term "demonstrating", when they should use terms like "predicting" or "inferring".

I. 566. The authors recommend that readers should only use one of the two current methods for measuring transcriptomes at the single cell level in order to "avoid method-induced biases". But that is exactly what they will be doing, providing a biased viewpoint of

the transcriptome that is dependent on the technology used. Please rephrase.

I. 651. "intestine" is listed, but since there are many different components of this tissue (stomach, duodenum, jejunum, ileum, cecum, etc.), please clarify. And if this included jejunum and ileum, it would be helpful if the authors indicated whether Peyer's patch was present/observed.

x. Several of the heat map and feature plot figures need scales. 2B, 2D, 2G, 2I, 3C, 3E

Dear reviewers,

We really appreciate all the valuable and constructive comments given by all of you to further improve our manuscript and affirm the findings. In this revision, we have thoroughly addressed all the comments and now provided a substantially revised manuscript. For your convenience, we highlighted a few major revisions here and a point-by-point response to all the individual comments are showed from the next page.

1. Encouraged by all your positive feedbacks on the first pig single cell atlas resource, we have generated a new, intuitive and interactive database based on the single cell RNA sequencing data generated. In the previous version, we created a static webpage which lacks the possibility of performing user-centered data visualization and analysis. To ensure that the resource will be broadly used and benefit the scientific community, we have generated a significantly improved atlas database using ShinyCell package. The URL of the pig single cell RNA atlas database (PCA) is <https://dreamapp.biomed.au.dk/pigatlas/>.

In the PCA database, we have provided all the QC plots. Most importantly, the new database provides many useful functions such as co-expression analysis, cell-type and gene expression comparisons. All high-resolution figures can be generated and freely download. Hope that you appreciate our efforts in making the data more accessible and useful. We will continue to integrate more single cell RNA sequencing data in the PCA database in the future.

2. We really appreciate the critical comments and valuable suggestions affirming and validating the EndMT ECs. We have reanalyzed our data with more stringent filtering criteria (removing fibroblasts and pericytes) for individual cells. In this revision, we provide a more systematic characterization of endothelial cell heterogeneity. While our data still support that there is EndMT phenotype in the adipose tissue ECs, we do indeed have less cells in the EndMT cluster. Most importantly, we have further conducted TGF β 2 induction experiment in cultured pig aorta endothelial cells and human umbilical cord vein endothelial cells. Collectively, we now provide more solid evidence supporting this small EC phenotype and its mechanism induced by the TGF β 2 signaling.
3. In the revised manuscript, all changes are highlighted with red font. All the NGS sequencing data, including the cultured ECs, have been shared in both CNGB and GEO data depository.

**On behalf of all authors,
Yonglun Luo, PhD, Associate Professor**

Reviewer: 1

Comments to the Author

In this submission, Wang and colleagues present single cell/nucleus RNA sequencing analysis of 20 pig organs, with the aim of generating an atlas of gene expression across tissues and cell types. They then focus on dissecting endothelial cell heterogeneity and transcription factor regulation of microglia identity. Although the objective of the study is important because of the relevance of the porcine model to studies of human disease and in xenotransplantation, there are several concerns that limit the utility of this data with respect to defining endothelial heterogeneity and superficiality of the analyses.

Author response: Thanks for all the constructive and valuable comments for clarifying all the concerns raised below. We are very grateful for all these important suggestions, which have now thoroughly addressed all the points. We have replenished some data analysis and performed more experimental validation for the EndMT in cultured ECs. Besides, to ensure that the pig single cell atlas can be more broadly used by the scientific community, and to facilitate the development of pigs for biomedical applications, we have created a more intuitive and user-friendly database (<https://dreamapp.biomed.au.dk/pigatlas/>). This open access database allows the visualizing and performing comparative gene expression analysis across tissues, cell types, cell stages etc. The revised manuscript has included all the valuable suggestions given by you and is now substantially improved.

Major comments:

1. Significant under-representation of endothelial cells in this dataset hampers investigation of their anatomic heterogeneity. Two other whole body single cell atlases (Tabula Sapiens and Tabula Muris) yielded substantively greater proportions of endothelial cells. It does not appear that sufficient endothelial cells were sequenced to provide an informative dataset to establish their transcriptional diversity. This was a limitation in Tabula Muris necessitating the endothelial-focused study by Kalucka et al.

Author response: We completely agree with the reviewer that, to systematically address the heterogeneity of endothelial cells in pig organs, it is needed to carry out another EC-focused study like the one (Kalucka et al.) that we had done previously. Unlike other model organisms, resources and understandings such as cell/tissue-specific gene expression, gene/genome annotation, cell type specific signatures, cell heterogeneity of the pig is lagging largely behind. This greatly hampers the use of pigs in biomedical research and applications. Only a few studies have been carried to understand the single cell gene expression and functions in pig tissues. In this study, we aim to provide the first pig single cell gene expression atlas across different organs. We selected the endothelial cells as a focus to further provide the first insight into EC heterogeneity in pigs. This is not the perfect strategy to fully address the EC heterogeneity, and we have highlighted this limitation in our revised

manuscript. We are currently carrying out EC-focused single cell RNA sequencing study in genetically modified pigs for xenotransplantation. We hope you agree with us that current manuscript provides the first valuable insights (though not systematically in all organs) into EC heterogeneity and functions.

2. In this atlas, the vast majority of endothelial cell events derive from adipose tissue, with very few from other highly vascularized organs which would be more relevant. Why is there a preponderance of adipose endothelial cells?

Author response: Sorry for not making this point clear in the manuscript. The preponderance of adipose endothelial cells captured in the adipose tissues was related to the protocol used, which was specifically developed and modified for enriching endothelial cells (ECs). We have highlighted this in the results section of the adipose ECs.

3. I could not determine from the methods what pipeline was used for cell type annotation? PECAM1 alone is not adequate to identify the endothelial lineage; it is expressed by other cell types, like monocytes, and it is lowly expressed by some endothelium, like liver sinusoids. Figure 3F shows that most of the other definitive markers of endothelial cells are in fact negative on the endothelial cells from all other organs. Could the authors please provide more information about how cell types were assigned?

Author response: For cell type annotation, we annotated all the major cell types based on the expression of canonical markers (fully list is provided in supplementary data S3) and differentially expressed genes for each cell types compared to all other cell types. There will be too many figures to be shown if we plot out all the marker genes. Instead, we provide all the canonical markers and top 50 significantly enriched marker for each cluster (cell type). We have now also generated the single cell RNA atlas database, which allows users to easily explore, plot, and download high resolution figures, as well as the QC plots.

We really appreciate the comments on the endothelial cell classification. We have included a more stringent marker-based selection criteria to extract the ECs from each tissue. Only cells positive for the pan-EC marker *PECAM1*, and negative for the epithelial cell marker *EPCAM*, negative for the immune cell marker *PTPRC*, negative for the fibroblast marker *COL1A1*, and negative for the pericyte marker *PDGFRB* are selected for EC heterogeneity analysis. The updated figure is now provided in Figure 3A and S3A. Also shown below for your reference.

Revised Figure 3A. Selection of ECs based on gene expression in individual cells

Revised Figure S3A. Dot plot of marker gene expression.

4. How can the authors be confident that the EndMT population is not simply contamination of fibroblasts or smooth muscle cells? There is a clear inverse correlation between PECAM1 expression and ACTA2/TAGLN, and population 9 in Figure 4C and population 4 in Figure 4G show low to negative PECAM1 and CDH5 expression compared with all other subsets. From this random snapshot in time, without a lineage tracer, it is difficult to say that cells co-expressing two markers (one low, one high) truly represent a transitional cell state. What evidence other than low PECAM1 expression can the authors provide to demonstrate that these cells are endothelial in origin?

Author response: We agree with the reviewer that simply based on the pseudo time trajectory analysis of the single cell transcriptome data could not provide highly

confident evidence that this cluster of cells are EndMT EC population. And we really appreciate the reviewer pointing out the potential contamination of fibroblasts or SMCs. To provide more solid evidence supporting that this cluster of ECs is EndMT, we have in this revision carried out substantial analyses as well as including TGFb2 induction experiments.

First, we have excluded all possible contaminated fibroblasts and pericytes by removing cells positive for *COL1A1* and *PDGFRB*. After more stringent filtering, the number of cells in the EndMT cluster was indeed smaller than the previous version, supporting that there was a fibroblasts contamination in the previous EndMT cluster. With the new parameter for EC selection, the expression trajectory of stimuli genes (*TGFb2* and *SNAI1*), endothelial cell markers (e.g., *PECAM1*, *VWF*, *CDH5*), and mesenchymal cell markers (e.g., *ACTA2*, *TAGLN*, *VIM*, *FABP4*) follows nicely the EndMT process (Figure 3F and 3G).

Second, we have reanalyzed the cultured EC single cell RNA sequencing data. Previously, we have only included the PECAM+ cells for analysis. In the revision, we have clustered all cells after removing low quality cells and doublets. We indeed identified a fibroblast cluster expressing *COL1A1* and *COL1A2*, in both cultured lung ECs and cultured aorta ECs. The presence of fibroblasts in the cultured ECs is expected as we isolate ECs from the tissue by perfusion of organs with enzymes (provided in the method section). Most importantly, we are able to identify a small cluster of ECs not expressing fibroblasts markers, but expressing mesenchymal cell marker *TAGLN* and *ACTA2* in PAECs. We defined this small cluster of EC as EndMT-like instead, as they weakly express EC genes. Based on the expression level of endothelial cell and mesenchymal cell markers.

Third, previous studies and our single cell analysis of the EndMT ECs in the adipose tissues suggest that TGFb2 is the key stimuli for the EndMT process. We have thus treated cultured pig aorta ECs and human umbilical cord vein endothelial cells (HUVEC) with TGFb2 and quantify the expression of ACTA2 and CD31 (PECAM1) by flowcytometry analysis. Our results showed that TGFb2 treatment for 5 days can significantly increase ACTA2 expression in the endothelial cells (Figure 4G-H, S4).

Collectively, we have now provided more solid evidence proving the EndMT phenotype and its regulation by TGFb2.

5. For the EndMT cells, could the authors also show expression of known regulators of this process, such as SNAI1, SNAI2, TWIST? Could they also please present a dot plot of co-expression of PECAM1-ACTA2, and other endothelial-mesenchymal genes?

Author Response: Thanks for this great suggestion. We have now included the co-expression dot plot of mesenchymal and endothelial cell markers. Our trajectory analysis suggests that SNAI1 is an early EndMT regulator, while SNAI2 is a late EndMT regulator according to the co-expression gene analysis. TWIST1 is an epithelial to mesenchymal transition regulator and was not enriched in our EndMT trajectory. These new results are provided in the revised Figure 3C, G, H.

6. The authors state that the data have been deposited into the CNGBdb, but I could not find the data in this database.

Author response: Thanks for your comments. The CNGB accession ID is CNP0002165 at the website of CNGBdb (data link: <https://db.cngb.org/search/project/CNP0002165/>).

We also shared our data on GEO data depository, accession numbers: GSE193975 (single cell and single nuclei RNA sequencing of pig tissues) and GSE196055 (single cell sequencing of cultured ECs).

7. Ethics statement is a bit awkward: "All experimental procedures were conducted following the guidelines of the experimental animals." Can the authors reword this to more specifically indicate which animal welfare guidelines were followed?

Author response: Great thanks. We have now revised the ethics statement accordingly, with both animal welfare guidelines and ethical approval ID included.

8. The authors conclude their manuscript with a statement that essentially invalidates their approach: "when the number and composition of cell types are essential, we recommend using only one method...consistently to avoid method-induced biases." One could argue that the number and composition of cell type ARE essential in this study. Therefore, could the authors better justify their approach in light of this?

Author response: Thank you for pointing out this. The suggestion of using only one method is not well justified and we have revised this discussion part accordingly. More precisely, we should compare the number the composition of cell types between tissues those are sequenced with the same strategy.

Minor comments:

1. Page 5, line 161: “sing-cell” should be “single-cell”

Author response: Thanks for pointing this mistake. We have corrected it in the revision.

2. Page 9, line 319: Should be “co-expression” rather than “co-expressing”

Author response: This is corrected accordingly.

3. Page 14, line 491: I believe the authors mean attachment rather than “detachment.”

Author response: Thanks for your kind comment. It was actually indeed meant detachment for the leucocytes. We have checked the original reference. Under physiological conditions, the endothelial cells are trying to avoid the attachment of leucocytes to the blood vessels through the CD31 signaling pathway. While under pathological conditions such as apoptosis or EC inflammation/activation, the CD31-mediated detachment signaling was taken over by the activation of adherent molecule pathways such as P-selectin, ICAM-1, VCAM-1, thus leading to the attachment of leukocytes to ECs.

Reference:

1. Liu, L. and G.P. Shi, CD31: beyond a marker for endothelial cells. *Cardiovasc Res*, 2012. 94(1): p. 3-5.

2. Brown, S., et al., Apoptosis disables CD31-mediated cell detachment from phagocytes promoting binding and engulfment. *Nature*, 2002. 418(6894): p. 200-3.

3. Newman, P.J. and D.K. Newman, Signal transduction pathways mediated by PECAM-1: new roles for an old molecule in platelet and vascular cell biology. *Arterioscler Thromb Vasc Biol*, 2003. 23(6): p. 953-64.

4. Page 14, line 518: “In addition, pathological angiogenesis in the development of human disease such as cancer.” This is an incomplete sentence.

Author response: Thanks for pointing this. We have revised the sentence as “In addition, the growth of pathological angiogenesis in human diseases such as cancers highlights that the targeting this process should help to reduce both morbidity and mortality from carcinomas (Nishida et al., 2006).”

5. Page 20, line 718: “To further ensure the quality of the dataset.” This is an incomplete sentence.

Author response: The sentence has been corrected.

6. The authors note that HLA-DRA was expressed by a subset of endothelial cells (Figure 3C). Should not the gene name be SLA-DRA (Swine Leukocyte Antigen). Fei Wang, Done!

Author response: Thank you for pointing this mistake. We have corrected the gene name by SLA-DRA (Swine Leukocyte Antigen) in line 272, 485, and 1225.

7. ICAM2 is rather a better marker of endothelial cells than ICAM1, which is expressed by many other cell types (Figure 4E).

Author response: Thank you for the suggestion, we have now included ICAM2 for both Figure 4C and 4E.

8. Axes are very small and illegible in Figure 2.

Author response: We have revised the axes in Figure 2 to make it clearer and more readable.

Reviewer: 2

Comments to the Author

Wang et al generated a single cell transcriptome atlas of many pig tissues using both single cell and single nuclei methodology. After performing adequate QC of the data, including comparison of single cell and nuclei data for a few of the tissues, they identified multiple known cell types and identify novel endothelial cell states. After the initial broad atlas analysis, they focus on: Identifying multiple sub-states for endothelial cells, and validated the proposed endothelial-mesenchymal state in vitro. Then, cell-cell interaction analysis is performed for endothelial cells. Finally, they perform an evolutionary analysis of microglia, and apply gene regulatory network analysis to find that MEF2C in a conserved "region" of these cells. In order to provide our best feedback and help authors detect inconsistencies, we re-ran the analysis using their provided processed files; we appreciate that the authors provided this data in a timely manner. Overall, this is a well done atlas with interesting findings. It will be an useful resource for the research community.

Author response: We really thank the reviewer's positive comments and the reviewer's help with re-ran the analysis for providing us all the valuable feedback. We have been working with pig genetics and genetic engineering for more many years already. Our previous research in pigs were really limited by the lack of such gene expression resources. Encouraged by the reviewer, we have now generated a more intuitive and user-friendly database for the research community. It allows users to explore the data more easily and generated almost unlimited number of figures based on their own research interests and focuses. The URL of the updated pig single cell RNA atlas is <https://dreamapp.biomed.au.dk/pigatlas/>.

However, like many other single cell atlas studies, the manuscript is affected by perhaps too broad a scope, and lack of focus.

#MAJOR COMMENTS

1. By including the description of general findings (e.g., in X tissue, we found X cell types expressing Y and Z), the manuscript becomes long, hard to follow, and dilutes the main findings that are discussed later. I think this dilutes the impact of the later findings. This is a common issue with most single cell atlas publications, and I don't have a great solution other than perhaps making the first few sections shorter and with less detail.

Author response: Thanks for pointing out this. In the revision, we have tried to remove some of the general description for cell type specific markers. For a few of them, we prefer to keep them to make the result part easier for understanding and to follow. We have provided the complete list of conical markers and enrich markers in the supplementary dataset instead. Combined with the new atlas database, readers should be able to explore the gene expression in cell types and tissues more easily. This might not be the perfect solution yet, but we think that it has improved significantly.

2. Then, the two main in depth analyses (the endothelial cell heterogeneity, and the microglia evolution and GRN) are shown, which are great but seem disconnected. Perhaps by shortening the initial general description sections the manuscript will be more readable. Otherwise, provide rationale as to why those two were selected for this manuscript.

Response: Thanks for your comments and suggestions. We aim to provide the first pig single cell RNA atlas for the scientific society. As a demonstration of the huge resource generated and how can it be used to gain better insights into cell functions (endothelial cells) and molecular evolution (microglia), we have selected the endothelial cells and microglia as two examples. We have now shortened the initial general description sections of manuscript, which is modified in first general description part. In addition, we added the descriptive sentences of rationale between the endothelial cells heterogeneity and the microglia evolution and GRN. That is " Single-cell transcriptomic analysis not only provides good insights into the cellular heterogeneity and functional diversity of structural cell types across tissues, but it is a good way to uncover the similarities and divergences of cell types across species. In this study, we utilized the big data of single-cell transcriptome of pig to analyze the cross-tissue ECs heterogeneity and ECs conversion in adipose. We are also interested in the cross-species cell types, which are mainly focused on microglia in the brain across 13 species. "

3. Please deposit data (raw and processed files) in GEO and provide accession number, add statement in manuscript with this information.

Author Response: Thanks for your comments. We have uploaded the raw sequencing files and expression matrices to GEO and the accession number is GSE193975 (data link: <https://www.ncbi.nlm.nih.gov/geo/query/acc.cgi?acc=GSE193975>). The statement in manuscript with this information is in line xxx.

4. We could not find three of the samples mentioned in the metadata (lung samples with sample IDs "PG-LG-1", "PG-LG-2", and "PG-LG-3"). We found 219,140 cells after passing their QC filter whereas the paper mentions 222,526 cells (i.e. I have 98% of the cells they mention). Please correct this so that data and manuscript match.

Author response: Great thanks for re-ran and double checking the analysis and pointing out the missing lung samples. The snRNA-seq data for pig lung were used from public data (CNSA: <https://db.cngb.org/cnsa>; accession number CNP0001486). We carefully checked and included the reference (Zhang et al. A high-resolution cell atlas of the domestic pig lung and an online platform for exploring lung single-cell

data). The processed matrix files for 222,526 cells mentioned in our manuscript were provided in the PCA database.

5. Provide reasoning to combine single cell and single nuclei approaches in this analysis, other than "practicality and resource availability"

Author response: We have added a rationale description of why we used to both two methods and the limitations between two methods. "Both scRNA-seq and snRNA-seq techniques have been used for single cell transcriptome analysis, which has both technical strengths and limitations (Liu et al., 2019). The scRNA-seq is performed using freshly isolated single cells, thus capturing all transcripts in the cells by limited by the sample processing procedures. Single cell suspensions must be prepared from the tissues immediately for scRNA-seq. For snRNA-seq, tissues can be snap frozen after sampling and used for nuclei extraction, thus not limited by timing. We selected both scRNA-seq and snRNA-seq for reasons of practicality and resource availability. To compare the two methods, four pig tissues (liver, retina, lung, spleen) were analyzed with both scRNA-seq and snRNA-seq (Figure 1A)."

6. It is unclear how many animals were sampled, and which tissues/cells are from which animals, and how they may be from the same animal or not. The metadata should include this, and include a table with this data (can be in supplement). Batch effects from sampling different animals can result in false "cell states" in clustering analysis (e.g. if all cells from a "novel state" come from a single animal), this affects the the conclusion of existence of Endothelial-mesenchymal transition cells.

Response: Thanks for your comments. We have now included the pig information in the metadata. Most importantly, the adipose tissues were from the same pig. We have performed more stringent filtering and analysis to validate that the small cluster of ECs are EndMT phenotype.

7. p9, lines 316-318: It would be helpful to show co-expression in the single cell RNA-seq data in addition to IF images. When we filtered only the cells that were PECAM+/PTPRC-/EPCAM-, there was a cluster of cells that were ACTA2+/TAGLN+ and a cluster of cells that were CD68+/C1QB+. Is it possible that these cells are actually smooth muscle cells? Similarly, is it possible that the "immune-activated" endothelial cells are macrophages? Please provide evidence using the scSeq data that these cells are distinct from smooth muscle and macrophages, respectively. For example, comparing the clusters against existing cell type databases (rather than using manually picked markers) may add further evidence of correct identification of cell clusters/types.

Author response: Thanks for the suggestion of shoring the co-expression in the single cell RNA-seq data. As the current figure is already highly compact, we have generated the single cell RNA atlas database using ShinyCell. This database allows

users to perform co-expression visualization and analysis for any pair of genes. For example, co-expression of PECAM1 and FABP4 in endothelial cells shown below.

Co-expression of FABP4 and PECAM1 in endothelial cells:

We fully understand the reviewer's concern that the two small clusters of ECs defined as immune-activated and EndMT could be macrophages and SMC respectively. We also thank the great suggestions for comparing to other external databased. One challenge for current resource of pig single cell gene expression is lack of such databases as in mouse and in human. Instead, we have taken more strict filtering criteria and conducted more experiment to provide more evidence supporting that these clusters of ECs are EndMT. In the revision (Figure 3), we have excluded possible contaminant of fibroblasts (*COL1A1*) and pericytes (*PDGFRB*). The immune-activated ECs are very unlikely to be macrophages, as we have removed all CD31+CD45+ cells and all macrophages are positive for CD45. Besides, we have also excluded the possibility that these might be doublet of EC and macrophages (Figure S3). In our previous studies of human and mouse ECs (i.e., Rohlenova K., et al. 2020 Cell Metabolism 31:862-877; Kalucka J., et al. 2020 Cell. 180, 764-779), this immune active EC phenotype was found in both health and pathological tissue, corroborating the immune modulating functions of ECs.

After stringent filtering, we have removed the contaminating fibroblasts in the EndMT EC phenotype. The EndMT is a phenotype describing the transition from endothelial cells toward mesenchymal cells. We have included pseudo time trajectory analysis, and our results suggest that the small cluster of EndMT ECs is undergoing a clear transition from ECs to mesenchymal cells (Figure 3F, G). The EndMT regulators TGFB2 and SNAI1 are enriched as the stimuli for the whole process, followed by the transition of EC and MSC marker genes.

To provide more solid evidence supporting the EndMT phenotype and TGFβ2 signaling pathway in driving this process, we induced cultured pig aorta ECs and human HUVECs with TGFβ2. Our results (Figure 4G, H) showed that significantly increase ACTA2 expression in the cultured ECs.

8. p10, lines 333-339: It does not seem we have access to the cultured primary ECs in the processed data. Please make sure raw and processed files are available in the deposited data.

Response: Thanks for your comments. We have uploaded the cultured primary ECs on GEO database with the number of GSE196055 (<https://www.ncbi.nlm.nih.gov/geo/query/acc.cgi?acc=GSE196055>).

9. p11, line 375: cell communication analysis does not prove that the cell-cell interaction is actually happening, it only suggests hypotheses to directly test; please reflect this in the writing, for example: "The analysis suggests that the VEGF signaling pathway is used by most renal cells for intercellular communication"

Response: Thanks for the suggestion. We have revised the correct wording accordingly.

#MINOR COMMENTS

1. p3 line 75: Merge two sentences for conciseness: However, several species-specific cellular and molecular differences between pigs and humans exist; for example, pluripotent progression and metabolic transition were found to be different using single cell RNA-sequencing (scRNA-seq) (Liu et al., 2021).

Response: The two sentences have been revised.

2. p3 line 87: Make sentence clearer, no need to list all species: "Pioneering work has been completed for most model animals and humans (References)"

Response: The sentence has been revised to make it clearer.

3. p4 line 114: change heading to "single-cell and single-nuclei RNA sequencing of four pig tissues highly correlates in common cell types"

Response: Great thanks for the suggestion. The heading has been revised accordingly.

4. p5 line 152: "also identified from the two different methods"

Response: This sentence has been revised.

5. p7 line 254: remove "much" -> have been extensively studied...

Response: We have removed "much" in this sentence in line.

6. p12 line 419: I think you mean "inference" rather than "interference"

Response: Thanks for pointing this mistake. We have corrected it in the revision.

7. Figure 1 inset (4) of Bioinformatic analysis says "virus receptor" and has a little diagram, but this was not included in this study, recommend removing.

Response: Great thanks for pointing out his mistake. We have removed the "virus receptor" and modified it in Figure 1B.

8. Figure 2 panel B: It appears that the heat map data (here and in other heat maps too) has been blurred during figure generation. One possible cause is the file format used to save, which sometimes causes this in heat maps. Please correct.

Response: Thanks for your suggestions. We have replaced this heat map with clearer one in the revised version.

Reviewer: 3

Comments to the Author

General comments

This manuscript reports single cell RNA seq data from approximately 20 adult tissues from the pig, with about 2/3 of the data coming from single cell and 1/3 coming from single nuclei technologies. This is an important resource for the scientific community interested in this species as a model for human, as well as investigations into the pig itself. The authors document similarities and differences between tissues at the single cell level and relationships at the transcriptome level across each tissue. They use as an example retina and kidney and document and verify these expression patterns using protein-based technologies. They then focus on endothelial cells and they provide some evidence for an intermediate stage cell type that is transitioning from an endothelial cell type to a mesenchymal cell type. They compare interacting pathways between human and pig for liver, heart and kidney and then finish the manuscript with a comparison across many species for the differences in gene expression in regulatory pathways present in microglia. The data analysis appeared sound, but additional details on the methodology are needed. Most importantly, they need to include more information on integration of the data, including normalization and scaling. This is essential for any single-cell data but especially important for projects integrating scRNA-seq and snRNA-seq. For the identification of cell clusters, they should explicitly state how many principal components were included and the resolution used for clustering.

While there are a number of places in the manuscript that should be revised, overall, the paper it does a very good job of describing a complex and deep data set that will be of value to many working in this species.

The major concerns are described in detail below, but include the low evidence for a clear transitioning population of EC to M cells, and the apparently simplistic description of the CellChat results. It would strengthen both of these results if at least some component was further documented or validated through statistical analysis or protein-level experimentation.

Author response: Thank you for all positive and constructive comments on the study and values that the study could bring to scientific society. In the revision, we have considered all the critical comments raised the reviewer and addressed them carefully and thoroughly. We have included more methodological details in the revised manuscript. For single cell RNA sequencing, most of the QC plots are important to be fully shown in the supplementary figures. To overcome this problem, we have generated a more intuitive database (<https://dreamapp.biomed.au.dk/pigatlas/>) to enable both exploring all the QC plots. Most importantly, the new database allows users to perform co-expression analyses.

In this revision, we have also performed more in-depth and introduced more stringent filtering steps to provide more solid evidence supporting the EndMT ECs. Furthermore, based on the finding of TGFb2 in driving the EndMT process, we performed TGFb2 induction experiments using cultured pig aorta endothelial cells and human umbilical cord vein endothelial cells. Our results collectively support the existence of EndMT MCs and validate that TGFb2 is the inducer of this process.

We have thoroughly addressed all the comments and revised our manuscript accordingly. A point-by-point response is provided below.

Specific comments

1. line. 131: The filter for removing cells only if they have $\geq 30\%$ mitochondrial transcripts is very high, and is concerning. 30% is a very high cutoff for mitochondrial reads for all tissues. There is precedence in using 30% for high energy tissues such as heart. However, the authors should explain why this metric was chosen, and if it is appropriate to use, for all tissues. The authors should also state what proportion of cells are in the dataset with $>10\text{-}20\%$ mito, which is more often the cutoff used.

Author response: Thanks for your comments. According to your suggestions, we have checked the percentage of cells that would be filtered out using 20% mitochondrial genes as cutoff. We found that only 4.9% cells were removed for the whole cells in our current pig dataset when we used 20% mitochondrial genes as cutoff. Besides, the boxplots below showed that most of the cells in each tissue and cell type can be retained using 20% mitochondrial genes as cutoff (under the red dashed line). These plots can be generated from the new PCA database. Therefore, we prefer to choose a relatively loose threshold (30% mitochondrial genes) in our dataset to ensure that more rare cell types are included while considering that the core findings can still be well supported. Noted that, samples analyzed with snRNA-seq have very low MT percentage.

MT% across tissues

MT% in cell types

Metadata

Platform	Tissue	Total cell number	<20% MT cell number	<20% MT cell proportion	The number of cells filtered out using 20% mitochondrial genes as cutoff
DNBelab C4	AreaPostrema	5109	5105	99.92170679	4
DNBelab C4	Cerebellum	10225	10017	97.96577017	208
DNBelab C4	Heart	3220	3100	96.27329193	120
DNBelab C4	Kidney	4255	4243	99.71797885	12
DNBelab C4	Liver	7395	7375	99.72954699	20
DNBelab C4	Lung	13575	13569	99.9558011	6
DNBelab C4	OVoLT	7739	7732	99.90954904	7
DNBelab C4	Retina	614	614	100	0
DNBelab C4	Spleen	3125	3117	99.744	8
DNBelab C4	SubfornicalOrgan	1527	1484	97.18402096	43
10X	Adipose-S	12814	11694	91.25955986	1120
10X	Adipose-V	16417	14492	88.27434976	1925
10X	Brain	15794	14492	91.75636318	1302
10X	FrontalLobe	8812	8812	100	0
10X	Hypothalamus	7302	7302	100	0
10X	Intestine	10518	8446	80.30043735	2072
10X	Liver	25493	24652	96.70105519	841
10X	Lung	12150	11930	98.18930041	220
10X	OccipitalLobe	6829	6829	100	0
10X	PBMC	10090	9886	97.97819623	204
10X	ParietalLobe	6162	6162	100	0
10X	Retina	11550	9098	78.77056277	2452
10X	Spleen	18666	18316	98.12493303	350
10X	TemporalLobe	3145	3145	100	0
Total		222526	211612		10914 4.90%

2. lines. 141-142: The data in S2 should be formatted to show a direct comparison between the two sequencing methods.

Author response: Thanks for the suggestion. We have revised Supplementary Data S2 accordingly.

3. lines. 145-146: It is difficult to see the different groups of cells sequenced in the tSNE plots. This is important to see how the data is impacted by sequencing method. For clarity, it might be better to show each group in its own tSNE plot.

Author response: We have re-organized the figures to show each group in its own tSNE plot in Figure S1.

Spleen:

Liver:

Lung:

Retina:

4. lines. 165-166: Figures 1D & 1E show the same data.

Response: Thanks for pointing out this problem. We have removed Figure 1D in the revised figure.

5. lines. 269-276: They suggest identification of “immunomodulating ECs” in their EC population given expression of several macrophage markers. Because in line 260 they indicate immune cells can express PECAM1, and the filter here was only to remove cells that express PTPRC, it is not at all clear to me that they actually excluded all macrophages from the EC pool (is it possible there are PTPRC- low or -negative macrophages?). If not, then this is not evidence of such an immunomodulating EC. Additionally, the authors should explain what expression metric was used and if individual cells were screened or if they applied the metrics to clustered cells. If it was not individual cells, there could be non-ECs included in the EC groups, which could impact downstream findings of transitioning cells.

Author response: Thank you for pointing out this. This a small EC phenotype that shares several functions with macrophage and have active immune modulating functions. We have characterized this EC phenotype in our previous single endothelial cell RNA sequencing studies (Jermaine Goveia, et al., 2020, Cancer Cell 37:21-36). And a recent review by Peter Carmeliet has specifically discussed this immunomodulating ECs (Nature Reviews Immunology 2022, March 14). In our data processing steps, we have screened the expression of individual cells, which clears out the possibility of contaminating macrophages.

6. lines. 338-348: I do not find these data to be highly convincing that they have clustered a transitioning cell type. They report in Figure 4C, E and describe that the pseudotime analysis shows a EMT using a set of genes that the trend line is almost completely flat for most genes. Only PECAM1/CDH5 shown any major expression changes across the x axis. The only evidence they have that there is a EMT cluster is that there are a few- very few- cells in black in 4E that are positive for PECAM1 (and even fewer for CDH5) and M markers such as ACTA2, etc. This is reflected in the very low numbers of cells reported in 4C that express the E markers in cluster 9, especially for CDH5. Similar complaints can be lodged for the work in aorta (Figure 4G), where PECAM1 and CDH5 expression is virtually non-existent in cluster 4. Is there any statistical test that can be run to document a transition is occurring, or to exclude the possibility that these cells are contaminants?

Author response: Great thanks for the suggestion and comments pointing the possibility of potential contaminants of other cells. In our previous analysis, we have only cluster cells positive for PECAM1 and did not exclude fibroblasts based COL1A1/COL1A2. In the revision, to ensure that we have the full set of data, we have analysis all primary cultured cells from lung and aorta. The ECs were isolated from the lung and aorta using an enzyme-based perfusion protocol. We identified a fibroblast cluster in both cultured lung and aorta cells, expressing COL1A1 and COL1A2. Besides, we still identified a very small population of EndMT population (late stage) expressing higher level ACTA2 and TAGLN in cultured aorta cells, but not the cultured lung ECs. However, we cannot be sure if this cluster of late stage EndMT cells are derived from the ECs or not since our medium does not contain TGFb2. To further prove that the pig EndMT can be induced by TGFb2, as demonstrated in figure 3, we culture PAECs and HUVECs in medium with TGFb2

and measured ACTA2 and CD31 expression by FACS. Our results showed that TGFb2 can rapidly induced ACTA2 expression in EC (more than 6 folds at day 5), while the CD31 expression was slightly decreased although not significant.

7. lines. 354-363: please add a short description of the principle behind CellChat, as it is not clear what is being measured here. In line 353-355, you describe that you are looking at “communication between ECs and other cell types”, and then compared RNAseq data between pig and human to “gain better understanding of the cell-cell interactions in these three organs between pig and human”. What is a “shared cell type” (l. 359), as distinct from simply shared gene expression patterns? Perhaps the explanation of what constitutes a “shared cell type” would suffice, but there could be further clarification on why there was not more shared cell types identified between pig and human (5-6 shared between tissues seems low).

Author response: Thanks for your kind comments. We have included a brief description for CellChat. Briefly, CellChat is a tool that is able to quantitatively infer and analyze intercellular communication networks from single-cell RNA-sequencing (scRNA-seq) data. Its performance for inferring the interactions between different cell types is mainly based on the different expression level of gene pairs corresponding to known receptor and ligand proteins on cell surface in different cell types. In detail, firstly the authors manually construct a database of ligand-receptor interactions that comprehensively considers the known structural composition of ligand-receptor interactions, such as multimeric ligand-receptor complexes, soluble agonists and antagonists, as well as stimulatory and inhibitory membrane-bound co-receptors. Next, CellChat takes gene expression data from cells as the user input and models the probability of cell–cell communication by integrating gene expression with prior knowledge of the interactions between signaling ligands, receptors and their cofactors. Then CellChat infers cell-state specific signaling communications within a given scRNA-seq data using mass action models, along with differential expression analysis and statistical tests on cell groups, which can be both discrete states or continuous states along the pseudotime cell trajectory. CellChat predicts major signaling inputs and outputs for cells and how those cells and signals coordinate for functions using network analysis and pattern recognition approaches. Through manifold learning and quantitative contrasts, CellChat classifies signaling pathways and delineates conserved and context-specific pathways across different dataset (Jin et al., Nature Communications, 2021).

For “shared cell type” in our manuscript, it refers to the cell type that we identified in both human and pig dataset from the same organ. For example, we found the existence of the following 6 major cell types in both pig kidney and human kidney, including epithelial cells, podocytes, proximal tubule cells, collecting duct cells, endothelial cells, distal convoluted tubule cells. Although we believe that more other shared cell types are to be discovered between human and pigs, here we only report these shared cell types supported by solid evidence (high expression of classical

marker) considering the influence of sampling differences, batch effect and limited number of cells, etc. The term of “shared cell type” is a little misleading, which we have revised it according in the revision.

8. line. 365: Cellchat apparently equates finding evidence of expression of a gene expressing a known ligand in cell A with expression of a receptor for that ligand in cell type B as “communication” between these cell types. The authors here appear to believe this inference, and “investigated signaling interactions” across cell types and tissues, and compared the results for the two species. There are many caveats to this analysis, none of which the authors admit or attempt to validate in any way with other techniques (such as demonstrating the presence of the ligand at the receiving tissue, or even expression of the protein of either partner in the tissues). For example, many ligands are not secreted in concert with gene expression levels, due to posttranscriptional or post-translational regulation. Further, the lack of such inferences based on a lack of such co-occurrences does NOT demonstrate a difference between species. There may be other explanations, such as biases using human genome annotations for the pig. The authors should describe the fairly large assumptions in these analyses, as well as at least some actual validations of activity of the inferred pathways. There was quite a bit of divergence in the results between cell signaling in pigs and human. Would the authors expect to see more conserved patterns between the two species?

Author response: Thanks for pointing out this. Although the CellChat and a few other similar tools developed for inferring cell-cell have been broadly used for single cell RNA sequencing data analysis, it is indeed that the information provided by such analysis will only provide a suggestive interaction between the different cell types due to the different caveats as already highlighted by the reviewer. Despite these caveats, these tools have been proven highly useful for providing the first insight to aid the identification of novel cell-cell interactions. Since this study was more focusing on providing the first pig single cell RNA atlas resource for the scientific society, we have in the revision highlighted these limitations and weakening the conclusions from the ligand-receptor pairs analysis.

9. line. 419: As above, please describe in more detail the method that you are using for GRN analysis, as there are several methods, rather than simply say you ran an analysis using GENIE3. (and “interfering“ presumably you mean “inferring“?). In this paragraph, the authors also use the term “demonstrating”, when they should use terms like “predicting” or “inferring”.

Response: Thanks for pointing this. We have included more detail for GRN analysis and several other data processing and analyses steps in the revised manuscript. We have also correct the terms in interpretating the results as suggested.

10. line. 566: The authors recommend that readers should only use one of the two current methods for measuring transcriptomes at the single cell level in order to “avoid method-induced biases”. But that is exactly what they will be doing, providing

a biased viewpoint of the transcriptome that is dependent on the technology used. Please rephrase.

Author response: Thanks for the suggestion. We have rephrase this sentence in the discussion part accordingly.

11. line. 651: “intestine” is listed, but since there are many different components of this tissue (stomach, duodenum, jejunum, ileum, cecum, etc.), please clarify. And if this included jejunum and ileum, it would be helpful if the authors indicated whether Peyer’s patch was present/observed.

Response: Thanks for your suggestions. We have checked our experimental records. The cells isolated from intestine were mainly original from small intestine part. We totally separated three small branches of small intestine of pig, and dissociated the cells for preparing single-cell suspension. The regions of intestine that we dissociated were showed below.

12. Several of the heat map and feature plot figures need scales. 2B, 2D, 2G, 2I, 3C, 3E

Response: Thanks for the question. We have checked our figures and provided the scales for Figure 2B, 2D, 2G, 2I, 3C, 3E.

REVIEWERS' COMMENTS

Reviewer #1 (Remarks to the Author):

In this revision and the point-by-point response, the authors have thoroughly and thoughtfully addressed my concerns.

I have no further recommendations.

Reviewer #2 (Remarks to the Author):

The authors have addressed my concerns and made significant efforts to improve the reproducibility of the data. I have no further comments.

Reviewer #3 (Remarks to the Author):

Specific comments on their responses to my review comments:

1. The addition of their new shiny tool to increase data availability is excellent. The explanation of relatively high Mito threshold is acceptable, and probably needed with so many different tissues and data types.
3. The authors split out the sc and sn data, which really helps you to see the different cell populations captured by each technology. The differences in cell capture are quite striking.
5. Not strongly convincing, but acceptable.
6. The new data is useful.
7. Acceptable.
8. Not strongly convincing, but acceptable.
12. The authors provided the requested scales but "Low – High" while informative, is not very precise. It would be better to have the actual value in these scales

Reviewer #1 (Remarks to the Author):

In this revision and the point-by-point response, the authors have thoroughly and thoughtfully addressed my concerns.

I have no further recommendations.

Author response: We are glad to hear that the reviewer agrees with us that the revision has thoroughly addressed all the concerns. We really appreciate all the valuable and constructive suggestions given by the reviewer for the improvement of my study and manuscript.

Reviewer #2 (Remarks to the Author):

The authors have addressed my concerns and made significant efforts to improve the reproducibility of the data. I have no further comments.

Author response: Thank you again for all the valuable and constructive suggestions for further improving my study and manuscript. We are happy to hear that the reviewer appreciates our effort to make the data reproducible and easily access by the scientific community.

Reviewer #3 (Remarks to the Author):

Specific comments on their responses to my review comments:

1. The addition of their new shiny tool to increase data availability is excellent. The explanation of relatively high Mito threshold is acceptable, and probably needed with so many different tissues and data types.

Author response: We are happy to hear that the reviewer appreciates our effort to make the data available through many possible ways. In addition to the shiny tool, we have now also collaborated with EBI to make the data available through the EBI single cell expression atlas. Thanks again for the comment on the MIT threshold.

3. The authors split out the sc and sn data, which really helps you to see the different cell populations captured by each technology. The differences in cell capture are quite striking.

Author response: Indeed, as we have pointed out in the discussion as well, the fractions of cells captured by the scRNA-seq and snRNA-seq method are quite strikingly different. However, this method-induced cell fraction variation has been well recognized.

5. Not strongly convincing, but acceptable.

Author response: We are happy to hear that this concern has been addressed.

6. The new data is useful.

Author response: We are happy to hear that the new data provides further useful evidence to support our finding.

7. Acceptable.

Author response: We are happy to hear that this concern has been addressed.

8. Not strongly convincing, but acceptable.

Author response: We are happy to hear that this concern has been addressed.

12. The authors provided the requested scales but “Low – High” while informative, is not very precise. It would be better to have the actual value in these scales.

Author response: Thanks for the suggestions. In the revised figures, we have used the actual values instead of scales. In the interactive pig atlas, we have provided two scales both actual value and scale ones.